# PHYCO: PHYSICS-CONSISTENT LEARNING OF IMPLICIT CONSTITUTIVE LAWS FROM DYNAMIC MONOCULAR OBSERVATIONS OF GAUSSIAN SPLATTING

## ABSTRACT

We present PHYCO, a framework for learning implicit constitutive laws from **monocular dynamic observations** of Gaussian splatting. Existing implicit methods often suffer from local minima under noisy supervision and lack physical interpretability, while explicit approaches rely on predefined constitutive equations, limiting generalizability. To address these issues, our framework, PHYCO, introduces two key innovations. First, **initializing from a static multi-view scan, we propose** *Edge-Aware Depth Consensus Anchors* **to establish robust geometric constraints from subsequent monocular dynamic observations**, circumventing unreliable pixel-level supervision. Second, a *Multi-Hypothesis Physics Verifier* integrates classical constitutive models as differentiable hypotheses, providing strong physical priors to regularize the optimization while preserving the flexibility of implicit modeling. This unified approach ensures physical plausibility without sacrificing generality. Extensive experiments on synthetic, real-to-sim, and real-world datasets demonstrate that PHYCO significantly outperforms existing methods, achieving state-of-the-art performance in learning accurate and generalizable physical dynamics from monocular videos.

## 1 INTRODUCTION

Understanding the intrinsic dynamics of objects is crucial for spatial intelligence and its applications, which require accurate digital modeling, interaction, and manipulation, following the physical laws (Yin et al., 2021; Juarez et al., 2021; Billard & Kragic, 2019; Nair et al., 2022). While humans can effortlessly infer basic physical properties from videos (e.g., bouncing balls or viscous fluid flow), extracting precise physical explanations from visual signals remains an open challenge.

Prior works have utilized AI models to achieve the goal of understanding the intrinsic dynamics (Chen et al., 2022; Jiang et al., 2024; Huang et al., 2020; 2024; Liu et al., 2024b). A common approach involves employing differentiable physics simulators (Dubied et al., 2022; Xue et al., 2023) to obtain object motion information, followed by using differentiable renderers (Mildenhall et al., 2021; Kerbl et al., 2023) to generate images. Regarding modeling of material constitutive laws, these approaches fall into two paradigms: explicit and implicit parameterization.

Explicit modeling (Huang et al., 2024; Liu et al., 2024a; Li et al., 2023; Zhang et al., 2024) builds upon classical continuum mechanics, constructing differential equation systems with predefined constitutive models (Drucker & Prager, 1952; der Wissenschaften zu Göttingen, 1922) (e.g., Hyperelastic (Stomakhin et al., 2012)) and explicit physical parameters like Young's modulus and Poisson's ratio. Through differentiable simulators (Hu et al., 2018a; Jiang et al., 2016), these methods achieve pixel-level supervision. Although enabling visual alignment, their effectiveness critically depends on predefined constitutive models: (1) This severely limits generalization as manual model specification and fine parameter tuning are required for different materials (Li et al., 2023); (2) They struggle with complex real-world materials exhibiting deeply coupled physical properties (Liu et al., 2024a).

In contrast, implicit parameterization methods model constitutive relations through neural networks (Ma et al., 2023; Li et al., 2022). NeuMA (Cao et al., 2024) introduces the first method to align

implicit constitutive models with visual observations. However, the implicit modeling paradigm, while maintaining generalization ability, introduces significant interpretability issues (Raissi et al., 2019) and optimization challenges—models easily converge to suboptimal solutions when handling noisy and sparse supervision (especially monocular videos) or complex material behaviors (Wang et al., 2023).

These conflicting trade-offs between explicit and implicit material modeling methods lead to our core research proposition: *How to reliably learn intrinsic dynamics from monocular videos while preserving the generalization advantages of implicit modeling?*

To bridge these gaps, we propose PHYCO, a physics-consistent learning framework to unify visual-physical bidirectional alignment for learning implicit constitutive laws from monocular videos. **It is important to note that while we utilize a standard static orbital scan for geometric initialization (following protocols like SpringGaus (Zhong et al., 2024)), our core contribution lies in learning intrinsic physical dynamics purely from monocular video supervision, where 3D tracking and physical identification are most challenging.** Unlike prior works, our method introduces two key innovations: (1) The Edge-Aware Depth Consensus Anchors extract robust geometric constraints from sparse observations, avoiding color domain shift-induced failures; (2) A Multi-Hypothesis Physics Verifier dynamically injects physics priors by treating classical constitutive models as differentiable hypotheses, ensuring plausibility without sacrificing flexibility. Extensive experiments validate that PHYCO significantly outperforms existing methods in both synthetic and real-world scenarios, paving the way for generalizable physics learning from monocular videos.

## 2 RELATED WORKS

### 2.1 PHYSICS-GROUNDED DYNAMIC 3D GENERATION

Dynamic 3D generation aims to capture an object's motion over time. While traditional NeRF-based models (Park et al., 2021; Fang et al., 2022; Kaneko, 2024; Feng et al., 2024b) are limited by predefined material assumptions, recent Gaussian-based methods (Kerbl et al., 2023; Feng et al., 2024a; Tan et al., 2024) show significant progress. For instance, Spring-Gaus (Zhong et al., 2024) uses spring-mass systems for elastic reconstruction, but still relies on an explicit model.

Some works attempt to learn physical knowledge from diffusion models for dynamic 3D Gaussian Splatting (GS) generation (Zhang et al., 2024; Liu et al., 2024a; Huang et al., 2024; Lin et al., 2025). However, diffusion models inherently lack rigorous physics-based image synthesis capabilities (Croitoru et al., 2023; Poole et al., 2022), making their implicit physical priors unreliable for precise perception tasks (Li et al., 2024). Moreover, these methods often rely on explicit physical model specifications (e.g., rigid (Liu et al., 2024b)/elastic body (Zhong et al., 2024) assumptions), limiting generalizable modeling. NeuMA (Cao et al., 2024) pioneers the optimization of neural constitutive laws directly from observational images without specific predefined physical laws. Nevertheless, under sparse supervision (such as monocular supervision and low frame rate) and highly complex physical properties (Xu et al., 2015; Xu & Barbič, 2017; Feng et al., 2024a), single-modality visual optimization suffers from local minima. Our method significantly enhances optimization stability in complex material scenarios by leveraging reliable multi-modal cues from sparse supervision.

### 2.2 MATERIAL CONSTITUTIVE LAWS

In continuum mechanics, material constitutive laws (Arruda & Boyce, 1993; der Wissenschaften zu Göttingen, 1922; Chhabra & Patel, 2023) govern responses to deformation and external forces. Conventional approaches for learning material constitutive laws (Cai et al., 2024; Liu et al., 2024a) enforce explicit constitutive laws via predefined nonlinear polynomial bases (e.g., elastic (Fung, 1967) / plastic (Drucker & Prager, 1952) / fluid models (Chhabra & Patel, 2023) ) and optimize parameters like Young's modulus or Poisson's ratio under rendering-based supervision. While ensuring physical consistency, these methods require manual design of constitutive equation forms (Liu et al., 2025) and initial parameters, severely limiting generalizable modeling (Meng et al., 2025).

Recent advances explore implicit neural constitutive modeling (Raissi et al., 2019; Cai et al., 2021; Lu et al., 2021). NCLaw (Ma et al., 2023) pioneers hybrid NN-PDE (neural network and partial differentiable equations), yet relies on precise particle-level annotations. NeuMA (Cao et al., 2024) further incorporates low-rank adaptation(LoRA) (Hu et al., 2022) to align implicit laws with visual observations via differentiable rendering, without particle-level supervision. However, pure visual

supervision lacks physical interpretability (Aira et al., 2024), and sparse or low-quality observations often lead to optimization ambiguity — implicit laws, despite their generalization potential, struggle to converge to physically plausible solutions without prior guidance. Our work proposes a hybrid constitutive framework that introduces a physical prior knowledge repository to regularize implicit optimization while avoiding overfitting to specific explicit models. This approach synergizes the optimization stability of explicit laws with the generalization capabilities of implicit laws, maintaining physical plausibility and accuracy under sparse supervision.

## 3 METHOD

### 3.1 PROBLEM STATEMENT

Given static 3D Gaussian kernels (Kerbl et al., 2023) of an object $\mathcal{G}(i) = \{\mathbf{p}(i), \alpha(i), \mathbf{A}(i), \mathbf{c}(i)\}$, where $\mathbf{p}(i), \alpha(i), \mathbf{A}(i), \mathbf{c}(i)$ are the center, opacity, covariance matrix, and spherical harmonic coefficients of each gaussian kernel, and its corresponding monocular dynamic video $\{I_t\}_{t=1}^T$, we aim to learn implicit constitutive laws through a dynamical system $\mathcal{M}_\theta$ governed by elastodynamics (Fung, 1977):

$$\rho_0 \ddot{\phi} = \nabla \cdot \mathbf{P} + \rho_0 \mathbf{b}, \quad \mathbf{P} = \mathcal{E}(\mathbf{F}_e), \quad \mathbf{F}_e = \nabla \phi, \tag{1}$$

where $\mathbf{P}$ is the first Piola-Kirchhoff stress tensor, $\rho_0$ is the object density, and $\mathbf{b}$ is the body force. Here, $\phi$ denotes the deformation map, and $\ddot{\phi}$ is its acceleration. $\mathcal{E}$ is defined by the elastic constitutive law. We discretize Eq. (1) and obtain the dynamical system $\mathcal{M}_\theta$:

$$\mathbf{s}_{t+1} = \mathcal{M}_\theta(\mathbf{s}_t), \quad \forall t = 0, 1, \dots, T-1, \tag{2}$$

where the states for physical simulation at $t$-th time step $\mathbf{s}_t = \{\mathbf{x}_t, \mathbf{v}_t, \mathbf{F}_e^t\}$. $\mathbf{x}_t, \mathbf{v}_t, \mathbf{F}_e^t$ are the particle positions, velocities, and elastic deformation gradients, respectively. $\theta$ is the neural parameters in $\mathcal{M}$. We provide details on preprocessing the gaussian kernels to particles for simulation in App. I.

To align $\mathcal{M}_\theta$ with the observation $I_t$, we use a differentiable renderer $\mathcal{R}$ producing $\hat{I}_t = \mathcal{R}(\mathbf{s}_t; \mathbf{K}, \mathbf{Q})$, where $\mathbf{K}, \mathbf{Q}$ denotes the camera's intrinsic and extrinsic matrices. Relying solely on this rendering-based supervision, however, is insufficient to overcome the challenges posed by sparse and noisy monocular video. The inherent *geometric ambiguities* from the single viewpoint and *material ambiguities* in the dynamics make the optimization landscape intractable. To establish a robust learning pipeline that addresses these fundamental issues, we propose our novel framework, PHYCO, short for physics-consistent learning (see Fig. 1). PHYCO operates through three coordinated mechanisms: First, we fine-tune neural material laws via low-rank adaptation (LoRA), ensuring compatibility with PDE-based physical simulations while maintaining parameter efficiency. Second, geometric ambiguities are resolved through the edge-aware depth consensus anchor that jointly optimize global motion coherence and local edge-aligned features. Finally, the multi-hypothesis physics verifier eliminates material ambiguities by enforcing hypothesis-driven physical constraints during sparse-view optimization, balancing generalization with dynamical consistency.

Next, we will provide a detailed explanation of each component in our framework. Sec. 3.2 introduces the differentiable neural material constitutive laws and the LoRA finetuning process, which serve as the foundation for our optimization task. Sec. 3.3 presents our strategy, edge-aware depth consensus anchor, designed to address the challenges of color inconsistency and geometric ambiguity in single-view scenarios. In Sec. 3.4, to tackle the unreliability of visual signals, we introduce multi-hypothesis physics verifier, a regularization approach that incorporates physical prior knowledge without compromising the model's generalization capability. This method avoids the need for specifying any explicit parameters and prevents the model from converging to suboptimal solutions.

### 3.2 NEURAL MATERIAL CONSTITUTIVE LAWS

Our work adopts the same dynamical system $\mathcal{M}_\theta$ as NCLaw (Ma et al., 2023) for state transitions. $\mathcal{M}_\theta$ is composed of the neural elasticity law $\mathcal{E}_{\theta_e}$, explicit Euler method (Hu et al., 2018b; Sulsky et al., 1995), and neural plasticity law $\mathcal{P}_{\theta_p}$. We use the basic physical prior model $\mathcal{M}_0 = \{\mathcal{E}_0, \mathcal{P}_0\}$ provided by NCLaw for state transitions. To align the model with observations without compromising the model's fundamental capabilities, instead of training all parameters in $\mathcal{M}_0$, we use LoRA (Hu et al., 2022) for finetuning. Specifically, we have $\mathcal{M}_\theta = \{\mathcal{E}_{\theta_e}, \mathcal{P}_{\theta_p}\}$, where $\mathcal{E}_{\theta_e} = \mathcal{E}_0 + \Delta\mathcal{E}_{\theta_e}$ and $\mathcal{P}_{\theta_p} = \mathcal{P}_0 + \Delta\mathcal{P}_{\theta_p}$.

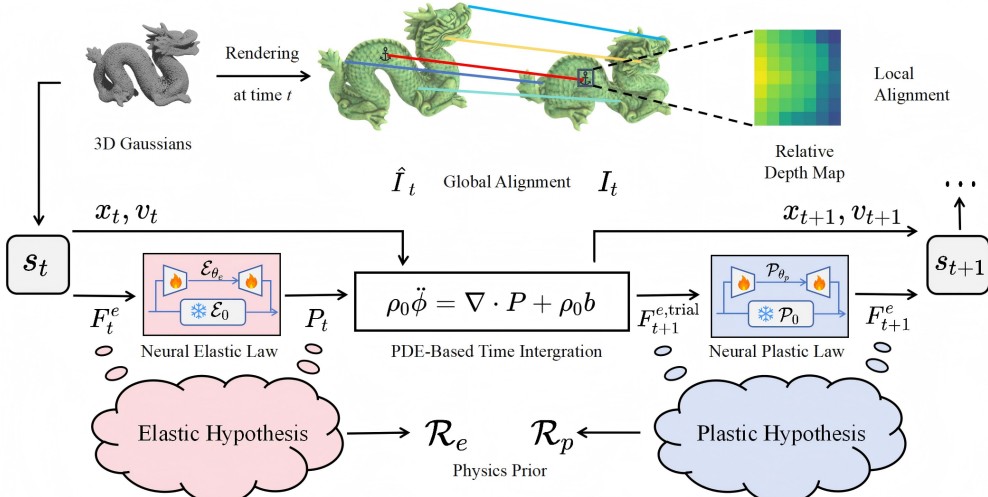

Figure 1: **Overview of our PHYCO framework.** PHYCO introduces three key technical components: (1) We employ **Low-Rank Adaptation** (LoRA) to fine-tune fundamental neural material laws while maintaining seamless integration with partial differential equation (PDE)-based physical simulation processes. (2) To address geometric ambiguities, we propose **Edge-Aware Depth Consensus Anchors** that resolve shape inconsistencies through joint alignment of global motion patterns and local geometric features. (3) For material ambiguity mitigation in sparse-view scenarios, we introduce the **Multi-Hypothesis Physics Verifier** that enforce physically consistent priors while preserving generalization capabilities through hypothesis-space constraints.

The dynamical system $\mathcal{M}_\theta$ advances physical states through three stages as shown in Alg. 1: (1) *Stress evaluation* via neural constitutive law $\mathcal{E}_{\theta_e}$ that computes first Piola-Kirchhoff stress $\mathbf{P}_t$ from elastic deformation gradient $\mathbf{F}_e^t$; (2) *Dynamics integration* where operator $\mathcal{I}$ implements semi-implicit Euler integration to update positions $\mathbf{x}_{t+1}$ and velocities $\mathbf{v}_{t+1}$ under inertia and body forces; (3) *Plasticity update* through network $\mathcal{P}_{\theta_p}$ that modifies $\mathbf{F}_e^{\text{trial}}$ to account for plastic deformation. The implementation details are provided in App. G.

---

**Algorithm 1** Time Stepping with Neural Constitutive Laws

---

**Require:** State $\mathbf{s}_t = \{\mathbf{x}_t, \mathbf{v}_t, \mathbf{F}_e^t\}$
**Ensure:** Next state $\mathbf{s}_{t+1}$
  **Stress Evaluation:**
  **for** each material point $i = 1$ to $N$ **do**
    $\mathbf{P}_t^{(i)} \leftarrow \mathcal{E}_{\theta_e}(\mathbf{F}_e^{t,(i)}; \theta_e)$
  **end for**
  **Euler Integration:**
  $\mathbf{x}_{t+1}, \mathbf{v}_{t+1}, \mathbf{F}_e^{\text{trial}} \leftarrow \mathcal{I}(\mathbf{x}_t, \mathbf{v}_t, \mathbf{P}_t)$
  **Plasticity Update:**
  **for** each material point $i = 1$ to $N$ **do**
    $\mathbf{F}_e^{t+1,(i)} \leftarrow \mathcal{P}_{\theta_p}(\mathbf{F}_e^{\text{trial},(i)}; \theta_p)$
  **end for**

---

### 3.3 EDGE-AWARE DEPTH CONSENSUS ANCHORS

Standard pixel-level color matching supervision used by methods like NeuMA (Cao et al., 2024) is unreliable for monocular video due to domain shifts and geometric ambiguities. To overcome this, we propose the Edge-Aware Depth Consensus Anchor. Instead of using color, our approach establishes robust geometric constraints by enforcing consensus between the rendered depth and depth maps generated by a pre-trained monocular depth estimator, focusing on stable object regions.

Given a rendered image $I_{\text{render}} \in \mathbb{R}^{H \times W \times 3}$, a rendered depth map $D_{\text{render}} \in \mathbb{R}^{H \times W}$ and a ground-truth (GT) image $I_{\text{gt}} \in \mathbb{R}^{H \times W \times 3}$, we first calculate the overall motion loss:

$$\mathcal{L}_{mask} = ||M_{render} - M_{gt}||_2^2, \tag{3}$$

where $M_{render}$ and $M_{gt}$ are the object region masks on the rendered and GT images, respectively. Mask information can offer a basic alignment, but it doesn't help solve color inconsistency and 3D geometric ambiguity.

We then utilize a pre-trained depth estimation network $\mathcal{D}$ (Yang et al., 2024) to generate relative depth maps $D_{gt}$, where the predicted depth values are geometrically consistent but lack absolute metric scale. A feature matching network $\mathcal{F}$ (Sarlin et al., 2020; DeTone et al., 2018) extracts $N$ 2D correspondence pairs $\{(p_i, q_i)\}_{i=1}^N$, where $p_i = (x_i, y_i)$ and $q_i = (x_i', y_i')$ denote matched coordinates in $I_{render}$ and $I_{gt}$, respectively. To mitigate depth estimation errors near object boundaries, we define the edge region $\mathcal{M}_{edge}$ through a single morphological opening operation:

$$\mathcal{M}_{edge} = \mathcal{M} \circ K_{r \times r}, \tag{4}$$

where $\circ$ denotes morphological opening (erosion followed by dilation) with a $r \times r$ rectangular kernel $K$. The stable interior region is correspondingly obtained as $\mathcal{M}_{stable} = \mathcal{M} \setminus \mathcal{M}_{edge}$.

For each correspondence pair $(p_i, q_i)$, we validate depth consensus in their local neighborhoods. Let $N_{p_i}$ and $N_{q_i}$ represent $k \times k$ regions centered at $p_i$ in $D_{render}$ and $q_i$ in $D_{gt}$, respectively. We compute the Spearman's rank correlation coefficient $\gamma_i$ (Sedgwick, 2014) for each pair $(p_i, q_i)$ between depth values in these neighborhoods:

$$\gamma_i = 1 - \frac{6 \sum_{j=1}^{k^2} (r_j - s_j)^2}{k^2 (k^4 - 1)}, \tag{5}$$

where $r_j$ and $s_j$ are the ranks of the $j$-th depth value in $N_{p_i}$ and $N_{q_i}$. A consensus indicator function $\phi(p_i, q_i)$ thresholds $\tau$:

$$\phi(p_i, q_i) = \mathbb{I}(\gamma_i > \tau), \tag{6}$$

with $\tau$ as the correlation threshold. Only pairs in $\mathcal{M}_{stable}$ satisfying $\phi(p_i, q_i) = 1$ are retained in the anchor set $\mathcal{A} = \{(p_i, q_i) \mid p_i \in \mathcal{M}_{stable} \wedge \phi(p_i, q_i) = 1\}$.

Our geometric alignment objective consists of two complementary components: a global alignment term that enforces overall consistency, and an anchor-level supervision term that preserves local geometric fidelity. The complete loss function is formulated as:

$$\mathcal{L}_{geo} = \underbrace{\lambda_1 \left\| P_{render} - P_{gt} \right\|_2}_{\text{global alignment}} + \lambda_2 \underbrace{\sum_{(p_i, q_i) \in \mathcal{A}} (-\gamma_i)}_{\text{anchor-level supervision}}, \tag{7}$$

where $P_{render}$ and $P_{gt}$ represent the matched point sets from rendered and ground truth images, respectively. The first term maintains global geometric consistency between the complete point sets, while the second term focuses on preserving precise local relationships through geometrically verified anchor pairs $\mathcal{A}$. The weighting factors $\lambda_1$ and $\lambda_2$ balance these complementary objectives. This hierarchical strategy combines broad-scale alignment with locally constrained refinement for robust optimization.

### 3.4 MULTI-HYPOTHESIS PHYSICS VERIFIER

To resolve material ambiguity in sparse-view settings, we design a physics verification process that evaluates candidate constitutive laws through parameter stability analysis. The key observation is that valid physical laws should produce consistent parameter estimates across deformation states, whereas invalid hypotheses lead to parameter divergence.

The elastic deformation gradient $\mathbf{F}_e^t \in \mathbb{R}^{N \times 3 \times 3}$ (spatial derivative of deformation map $\phi^t$) serves as input, while the outputs are physics residuals $\mathcal{R}_e, \mathcal{R}_p$ quantifying deviation from plausible laws. Various candidate laws $\mathcal{H}_e$ (elastic) and $\mathcal{H}_p$ (plastic) are defined in App. H (Chhabra & Patel, 2023; Drucker & Prager, 1952; Fung, 1967; der Wissenschaften zu Göttingen, 1922), serving as our constitutive hypotheses. These candidate laws are widely used (Meng et al., 2025; Liu et al., 2025) in the field of materials. For each candidate law, we use $\mathbf{F}_e^t[\mathcal{S}, :, :]$ to evaluate whether the implicit material models align with this law, where $\mathcal{S} \subset \{1, ..., N\}$ is a fixed subset of indices to reduce the calculation cost.

The Multi-Hypothesis Physics Verifier (Alg. 2) operates through three key phases to enforce physical consistency. First, it performs standard elastic-plastic simulation steps: (1) elastic stress prediction

---

**Algorithm 2** Multi-Hypothesis Physics Verifier

---

**Require:**
$\mathbf{F}_e^t \in \mathbb{R}^{N \times 3 \times 3}$, $\mathcal{H}_e = \{\mathcal{H}_e^k\}_{k=1}^K$, $\mathcal{H}_p = \{\mathcal{H}_p^m\}_{m=1}^M$, $\mathcal{E}_{\theta_e} : (\mathbf{F}_e^t \to \mathbf{P}^t)$, $\mathcal{P}_{\theta_p} : (\mathbf{F}_e^{\text{trial}} \to \mathbf{F}_e^{t+1})$, $\epsilon$,
$\mathcal{S} \subset \{1, ..., N\}$
**Ensure:**
$\mathbf{F}_e^{t+1}, \mathcal{R}_e^t, \mathcal{R}_p^t$
**Elastic Stress Prediction:** $\mathbf{P}^t \leftarrow \mathcal{E}_{\theta_e}(\mathbf{F}_e^t)$
**Eular Integration:** $\mathbf{F}_e^{\text{trial}} \leftarrow \mathcal{I}(\mathbf{P}^t)$
**Plasticity Correction:** $\mathbf{F}_e^{t+1} \leftarrow \mathcal{P}_{\theta_p}(\mathbf{F}_e^{\text{trial}})$
**Parameter Solving:**
$\mathbf{F}_e^{t,\mathcal{S}} \leftarrow \mathbf{F}_e^t[\mathcal{S}, :, :]$
$\mathbf{F}_e^{\text{trial},\mathcal{S}} \leftarrow \mathbf{F}_e^{\text{trial}}[\mathcal{S}, :, :]$
**for** $k = 1$ **to** $K$ **do**
    $\hat{\Theta}_e^k \leftarrow \underset{\Theta_e^k}{\arg\min} \|\mathcal{E}_{\theta_e}(\mathbf{F}_e^{t,\mathcal{S}}) - \mathcal{H}_e^k(\mathbf{F}_e^{t,\mathcal{S}}; \Theta_e^k)\|_F^2$
    $\omega_e^k \leftarrow \frac{1}{\text{Var}\{\hat{\Theta}_e^k\} + \epsilon}$
**end for**
**for** $m = 1$ **to** $M$ **do**
    $\hat{\Theta}_p^m \leftarrow \underset{\Theta_p^m}{\arg\min} \|\mathcal{P}_{\theta_p}(\mathbf{F}_e^{\text{trial},\mathcal{S}}) - \mathcal{H}_p^m(\mathbf{F}_e^{\text{trial},\mathcal{S}}; \Theta_p^m)\|_F^2$
    $\omega_p^m \leftarrow \frac{1}{\text{Var}\{\hat{\Theta}_p^m\} + \epsilon}$
**end for**
$\mathcal{R}_e^t \leftarrow \sum_{k=1}^K \omega_e^k \|\mathcal{E}_{\theta_e}(\mathbf{F}_e^t) - \mathcal{H}_e^k(\mathbf{F}_e^t; \mathbb{E}[\hat{\Theta}_e^k])\|_F^2$
$\mathcal{R}_p^t \leftarrow \sum_{m=1}^M \omega_p^m \|\mathcal{P}_{\theta_p}(\mathbf{F}_e^{\text{trial}}) - \mathcal{H}_p^m(\mathbf{F}_e^{\text{trial}}; \mathbb{E}[\hat{\Theta}_p^m])\|_F^2$

---

via $\mathcal{E}_{\theta_e}$, (2) Euler integration through $\mathcal{I}$, and (3) plasticity correction using $\mathcal{P}_{\theta_p}$. Next, the algorithm solves inverse problems to estimate explicit parameters $\Theta$ for each candidate law ($\mathcal{H}_e^k$, $\mathcal{H}_p^m$) that minimize the discrepancy with learned material responses on a sampled subset $\mathcal{S}$. The credibility weights $\omega$ are computed as inverse variance measures (with smoothing factor $\epsilon$), assigning higher confidence to laws with stable parameter estimates. Finally, physical residuals $\mathcal{R}_e$ and $\mathcal{R}_p$ penalize deviations from credible laws using weighted combinations of hypothesis deviations, where $\mathbb{E}[\hat{\Theta}]$ represents averaged stable parameters. These residuals provide physical priors for $\mathcal{E}_{\theta_e}$ and $\mathcal{P}_{\theta_p}$ respectively during optimization.

**Overall Optimization Objectives.** In summary, the overall optimization objectives of the neural elastic model $\mathcal{E}_{\theta_e}$ and the neural plasticity model $\mathcal{P}_{\theta_p}$ are

$$\mathcal{L}_e = \lambda_m \mathcal{L}_{mask} + \lambda_g \mathcal{L}_{geo} + \mathcal{R}_e, \tag{8}$$

$$\mathcal{L}_p = \lambda_m \mathcal{L}_{mask} + \lambda_g \mathcal{L}_{geo} + \mathcal{R}_p \tag{9}$$

respectively, where $\lambda_m$ and $\lambda_g$ are balance factors. We provide a theoretical analysis of the convergence properties of our optimization framework in App. J.

## 4 EXPERIMENTS

**Experimental Setup.** To comprehensively evaluate the superiority of our method, we conduct systematic validation across three data dimensions: *fully synthetic*, *real-to-sim*, and *real-world*.

We conduct all experiments on a single NVIDIA A800 80GB GPU. Our framework is computationally efficient, and a detailed analysis comparing its training time, inference speed, and memory usage against the baseline is provided in the App. F for interested readers.

For *synthetic* experiments, we utilize the NeuMA dataset (Cao et al., 2024) which provides benchmark videos with multiple physical material properties. However, its idealized color consistency assumption (where ground-truth videos achieve perfect pixel alignment with rendered sequences) fails to reflect prevalent color discrepancies in real physical scenarios. To address this limitation, we construct a more challenging synthetic benchmark containing six material types (elastomers, gels, rubber, plasticine, granular materials, and non-Newtonian fluids) across diverse object geometries (spheres, ducks, pawns, cats, fish, and bottles). Our enhanced benchmark is introduced in the App. A

To bridge the gap between synthetic and real-world scenarios, we introduce a novel *real-to-sim* dataset. This dataset is created by first capturing high-quality 3D Gaussian Splatting models of real objects, including *dragon*, *wolf*, and *pudding*. We then use these static models as initial states in a physics simulator to generate dynamic sequences with complex material properties. This setup provides ground-truth physics while retaining the geometric and appearance complexity of real objects.

For *real-world* validation, we adopt the SpringGaus dataset (Zhong et al., 2024) containing tri-view video sequences of four moving objects. Distinct from existing methods relying on multi-view supervision, we strictly constrain our approach to monocular video supervision, better aligning with practical application constraints. This setting significantly increases modeling difficulty but better demonstrates the method's practical value.

**Baseline Methods.** We evaluate our method under monocular video supervision, comparing against state-of-the-art approaches including NCLaw and NeuMA on synthetic data, while employing SpringGaus' original method and NeuMA migrated models for real-world validation. Our approach specifically addresses the challenging but practical monocular setting, in contrast to methods like GIC (Cai et al., 2024) and PAC-NeRF (Li et al., 2023) which require dense multi-view supervision as mandatory input. We further exclude approaches such as PhysDreamer (Zhang et al., 2024) and Physics3D (Liu et al., 2024a) from comparison since their reliance on predefined explicit constitutive models and diffusion guidance (Croitoru et al., 2023) fundamentally violates our general modeling assumptions of learning implicit constitutive laws directly from visual observations.

**Evaluation Metrics.** We follow previous work to evaluate the performance: (1) Chamfer Distance (Butt & Maragos, 1998; Erler et al., 2020) for geometric consistency, (2) SSIM (Wang et al., 2004) for structural similarity, (3) PSNR (Hore & Ziou, 2010) for pixel-level reconstruction accuracy, and (4) LPIPS (Zhang et al., 2018) for perceptual similarity.

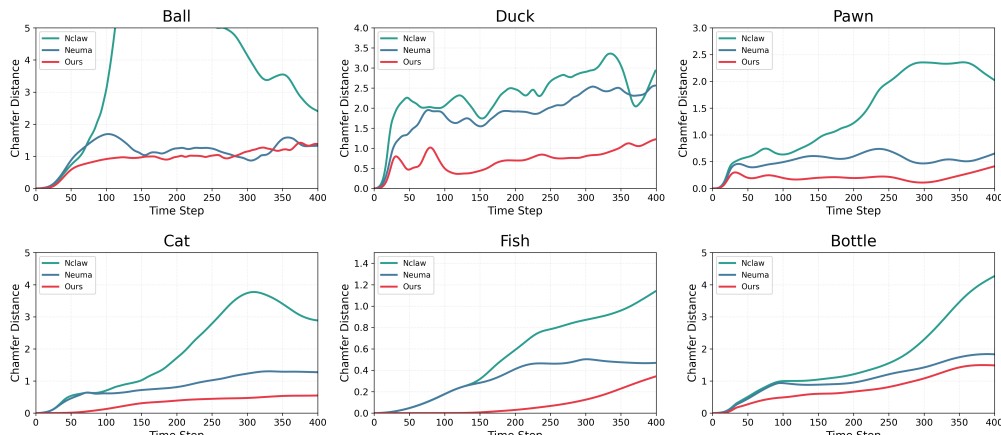

Figure 2: **Chamfer distance during physical simulation.** Our method consistently maintains lower Chamfer Distance than baseline methods throughout the physical simulation process, demonstrating that the learned implicit physical properties effectively represent the intrinsic dynamics of objects.

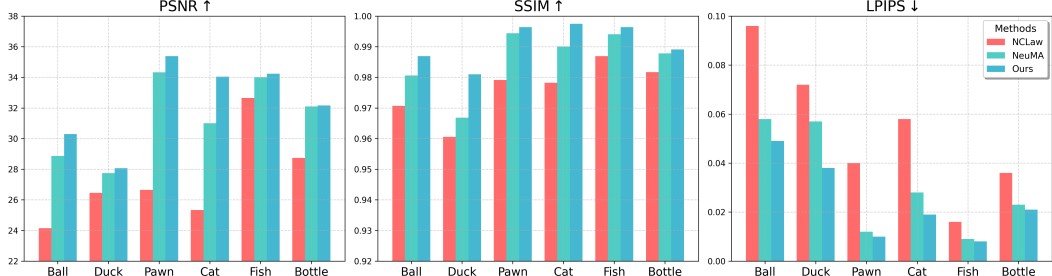

Figure 3: **Quantitative comparison on the synthesized dataset with rendering metrics.** Our method achieves superior performance across all three image quality metrics (PSNR, SSIM, and LPIPS) compared to baseline approaches, demonstrating that our rendered videos more accurately capture the intrinsic dynamics of physical objects.

## 4.1 EVALUATION ON SYNTHETIC DATASET

We evaluate physical simulation accuracy by computing Chamfer Distance (CD) between predicted and ground-truth particle positions in synthetic experiments as shown in Tab. 1. Results demonstrate that under the challenging benchmark with color inconsistency and sparse supervision signals, our method effectively captures implicit physical laws from visual data, achieving significantly lower CD values than baselines. This validates the method's effective modeling capability in complex observation conditions.

Table 1: **Quantitative comparison in the synthesized dataset in Chamfer distance.** We compare our method against baselines NCLaw (Ma et al., 2023) and NeuMA (Cao et al., 2024). Our method consistently achieves a 48% average lower Chamfer Distance (compared to ground-truth) than NeuMA across diverse object geometries and material properties, demonstrating its superior capability in learning intrinsic dynamics from monocular videos.

| Material Object | Elastomer Ball | Gel Duck | Rubber Pawn | Plasticine Cat | Granular Fish | Non-Newtonian Bottle | Average |
|---|---|---|---|---|---|---|---|
| NCLaw | 4.085 | 2.934 | 2.031 | 1.909 | 0.536 | 1.631 | 2.188 |
| NeuMA | 1.123 | 1.863 | 0.517 | 0.844 | 0.322 | 1.056 | 0.954 |
| Ours | **0.922** | **0.702** | **0.200** | **0.318** | **0.077** | **0.757** | **0.496** |

Fig. 2 illustrates temporal CD variations during physical simulation. Notably, our method maintains alignment with the GT throughout the simulation, while baselines gradually deviate from GT trajectories with increasing timesteps. This confirms the method's robustness in long-term physical evolution modeling.

Further quantitative comparisons on rendering metrics are provided in Fig. 3. Experiments show that our method accurately captures motion patterns despite increased material complexity, whereas baselines exhibit significant distortion under interference. This validates our method's capability in extracting essential physical laws from noisy observations, even without direct qualitative visualization in the main paper.

## 4.2 EVALUATION ON REAL-TO-SIM DATASET

We use our newly introduced *real-to-sim* dataset, consisting of *dragon*, *wolf*, and *pudding*, to assess the generalization capability of our method on complex geometries derived from real objects. Further details on the creation of our dataset are available in App. B. Tab. 2 shows the quantitative results, where PhyCo consistently outperforms the baselines.Fig. 4 provides a qualitative comparison on this dataset. Our method successfully learns plausible dynamics for complex objects like the *dragon*, *wolf* and *pudding*, generating renderings that are both physically consistent and visually aligned with the ground truth. In contrast, baseline methods struggle to capture the correct deformation, resulting in noticeable artifacts and unrealistic motion. This highlights our framework's superior ability to generalize to challenging, realistic scenarios.

Table 2: **Quantitative comparison on the real-to-sim dataset in Chamfer distance.** Our method achieves the lowest error, validating its ability to generalize learned physical laws to complex, real-world geometries.

| Object | Plasticine Dragon | Sand Wolf | Gel Pudding |
|---|---|---|---|
| NCLaw (Ma et al., 2023) | 25.021 | 49.420 | 38.149 |
| NeuMA (Cao et al., 2024) | 3.527 | 9.803 | 13.804 |
| Ours | **2.081** | **5.842** | **0.906** |

## 4.3 EVALUATION ON REAL-WORLD DATASET

To verify the generalization capability in real scenarios, we conduct monocular supervision experiments on the SpringGaus dataset (Zhong et al., 2024). Notably, while the original SpringGaus setup employs tri-view video supervision, our study strictly uses single-view videos as supervision signals. As shown in Fig. 5, under monocular supervision, PHYCO successfully disentangles implicit physical properties from observations and demonstrates strong generalization under strict monocular constraints.

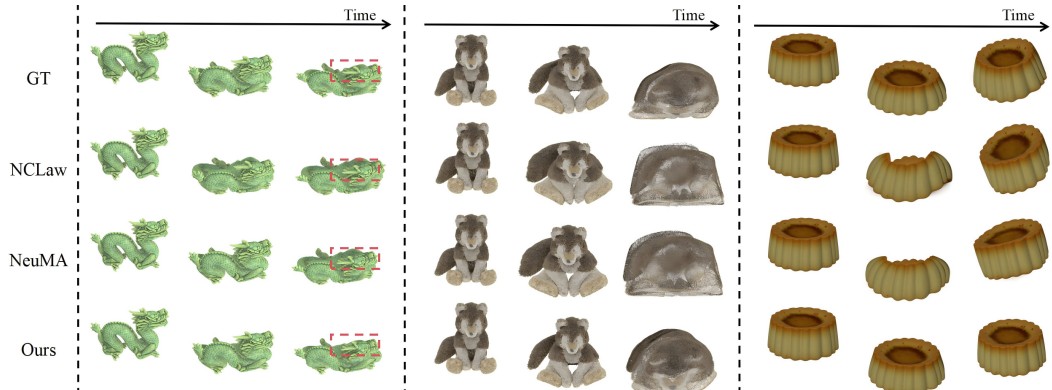

Figure 4: **Qualitative comparison on the real-to-sim dataset.** Our method accurately captures the complex dynamics of objects derived from real-world scans (e.g., *dragon*, *wolf*, *pudding*), producing physically plausible and visually superior results compared to the baselines.

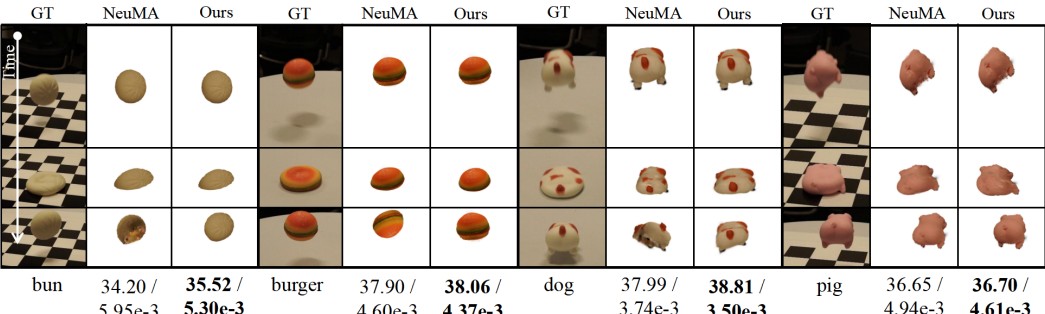

| bun | 34.20 / | **35.52 /** | burger | 37.90 / | **38.06 /** | dog | 37.99 / | **38.81 /** | pig | 36.65 / | **36.70 /** |
|-----|---------|-------------|--------|---------|-------------|-----|---------|-------------|-----|---------|-------------|
|     | 5.95e-3 | **5.30e-3** |        | 4.60e-3 | **4.37e-3** |     | 3.74e-3 | **3.50e-3** |     | 4.94e-3 | **4.61e-3** |

Figure 5: **Qualitative comparison on real-world dataset.** On real-world datasets, our method achieves superior rendered image quality using only monocular video supervision, while baseline approaches fail to reliably learn object physical properties under the same monocular supervision constraints. We also present quantitative results (PSNR / LPIPS) between predictions and observations (with background filtered) in the bottom row.

### 4.4 ABLATION AND GENERALIZATION STUDIES

To further validate our framework, we conduct comprehensive ablation and generalization studies, with full details provided in App. E and App. D. Our ablation analysis confirms that both the **Edge-Aware Depth Consensus Anchors** and the **Multi-Hypothesis Physics Verifier** are critical components, as removing either results in a significant performance drop. Furthermore, our generalization experiments demonstrate that the learned physical properties are transferable to novel multi-object interaction scenarios, indicating that our method successfully captures intrinsic material properties rather than overfitting to the training scenes.

## 5 CONCLUSION

We presented PHYCO, a framework for learning implicit constitutive laws from monocular videos through visual-physical bidirectional alignment. By integrating Edge-Aware Depth Consensus Anchors and a Multi-Hypothesis Physics Verifier, our method achieves stable optimization under sparse and noisy supervision while preserving physical interpretability. Quantitative results show significant improvements on synthetic data (48% lower Chamfer Distance than NeuMA), strong generalization on a challenging real-to-sim benchmark, and higher quality than other baselines in real-world monocular experiments. Future work may extend to dynamic multi-object interaction modeling or more real-world experiments.

### REPRODUCIBILITY STATEMENT

To ensure the reproducibility of our findings, we have included our core implementation and custom dataset in the supplementary materials. Specifically, the submitted supplementary ZIP file contains: (1) The source code for our PHYCO framework, including the implementation of the Edge-Aware

Depth Consensus Anchors and the Multi-Hypothesis Physics Verifier. (2) Our complete real-to-sim dataset, which includes the corresponding dynamic video sequences for *dragon*, *wolf*, and *pudding* assets. All necessary details required to run our experiments are documented in the appendix.

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

## A    DETAILS ON THE ENHANCED SYNTHETIC DATASET

In this section, we present technical details of our PHYCO-Synthetic benchmark for learning intrinsic object dynamics.

While NeuMA(Cao et al., 2024) provides a geometry-based synthetic dataset spanning elastomers to plastic bodies, its configurations exhibit three critical simplifications. Our benchmark introduces three critical improvements over prior synthetic datasets:

- **Dynamic Lighting Interference**: We incorporate randomized lighting interference in dynamic videos to explicitly break the color consistency between rendered static Gaussians and observed frames, addressing the unrealistic environmental uniformity in NeuMA. The training frame rate is reduced to practical 125/250 FPS (while retaining 2000 FPS raw data) to match real-world acquisition constraints.
- **Compound material modeling**: Materials are synthesized through compound constitutive laws combining a primary and an auxiliary physical effects, systematically reflecting the dominance-subordination relationships observed in real-world material behaviors, unlike NeuMA's oversimplified single-constitutive representations.
- **Practical Frame Rates**: NeuMA uses 1000/2000 FPS supervision videos, which are beyond practical acquisition capabilities. We adopt practical frame rates (125/250 FPS) for training while preserving full 2000 FPS data for completeness

This design ensures both physical fidelity and reproducibility while maintaining backward compatibility with existing methods.

The details are shown in the Tab. 3. And the impact of lighting interference is shown in Fig. 6

Table 3: **Details about our synthesized dataset.**

| Asset | Material | Step Size(s) | FPS(training) |
|--------|---------------|--------------|---------------|
| Ball | Elastomer | 1e-3 | 250 |
| Duck | Gel | 1e-3 | 250 |
| Pawn | Rubber | 5e-4 | 125 |
| Cat | Plasticine | 5e-4 | 125 |
| Fish | Granular | 5e-4 | 125 |
| Bottle | Non-Newtonian | 5e-4 | 125 |

## B    DETAILS ON THE REAL-TO-SIM DATASET

To further bridge the gap between synthetic benchmarks and real-world complexity, we curated a challenging *real-to-sim* dataset. The creation process begins by capturing high-fidelity 3D models of real objects—a dragon statue, a wolf figurine, and a pudding dessert—which are then reconstructed as high-quality 3D Gaussian Splatting scenes. These static reconstructions serve as the initial state for our physics simulations.

We then employ an MPM-based simulator to generate dynamic video sequences. By assigning distinct and complex material properties (e.g., plasticine, granular material, and gel) to these realistic geometries, we produce physically accurate ground-truth dynamics for objects with intricate shapes and textures. This dataset is crucial for evaluating a model's ability to generalize learned physical laws to the variety seen in real-world applications. The details for each asset are provided in Tab. 4.

## C    QUALITATIVE VISUALIZATION ON THE SYNTHETIC DATASET

In this section, we provide a qualitative comparison of our method against baselines on the purely synthetic dataset. It is worth noting that the synthetic data provides a relatively controlled and

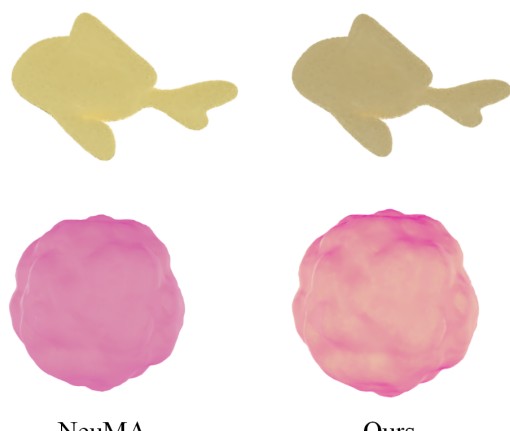

NeuMA         Ours

Figure 6: **We introduce randomized lighting perturbations to the rendered outputs, thereby increasing the challenge level for optimization tasks.**

Table 4: **Details of our real-to-sim dataset assets.** Each asset originates from a real-world object and is used to generate a dynamic sequence with specified material properties via MPM simulation.

| Asset | Material | Step Size(s) | FPS (training) |
|---|---|---|---|
| *dragon* | Plasticine | 1e-3 | 250 |
| *wolf* | Granular | 1e-3 | 250 |
| *pudding* | Gel | 1e-3 | 250 |

simplified environment (e.g., uniform backgrounds, less complex textures) compared to the real-to-sim and real-world datasets. Consequently, all methods are capable of achieving reasonably good performance, and the visual differences are not as pronounced.

However, as shown in Fig. 7, a closer inspection of the results for the **Cat** (Plasticine) and **Ball** (Elastomer) assets reveals the superiority of our approach. By zooming in, one can observe that our method, PHYCO, generates dynamic sequences with more plausible surface deformations and fewer visual artifacts. This demonstrates that even in simpler scenarios, our physics-regularized framework produces higher-fidelity results.

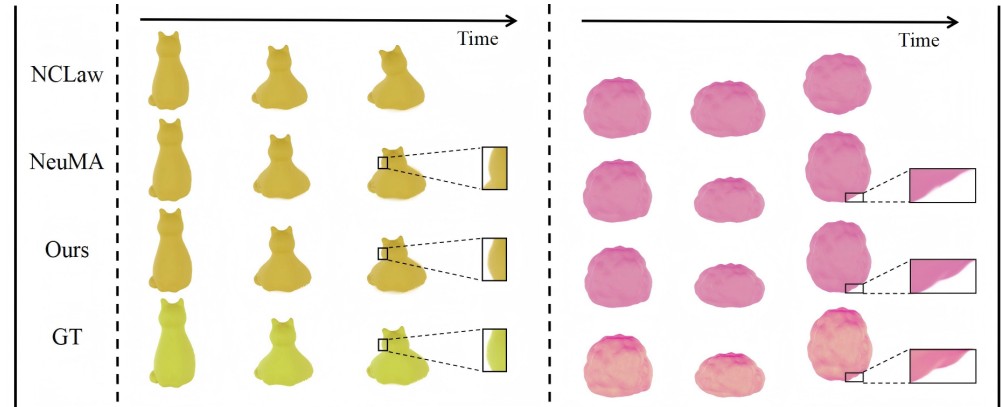

Figure 7: **Qualitative comparison on the synthetic dataset.** While the overall quality is comparable due to the simplicity of the task, a closer look at the Cat and Ball examples shows that our method produces results with higher physical fidelity and fewer artifacts than the baselines.

## D GENERALIZATION RESULTS

In this section, we demonstrate that the physical properties learned by our method can be transferred to novel objects and effectively support multi-object interaction rendering. As shown in Fig. 8, we apply distinct learned physical attributes to identical object instances, verifying that our implicitly acquired properties correctly manifest the materials' intrinsic dynamics.

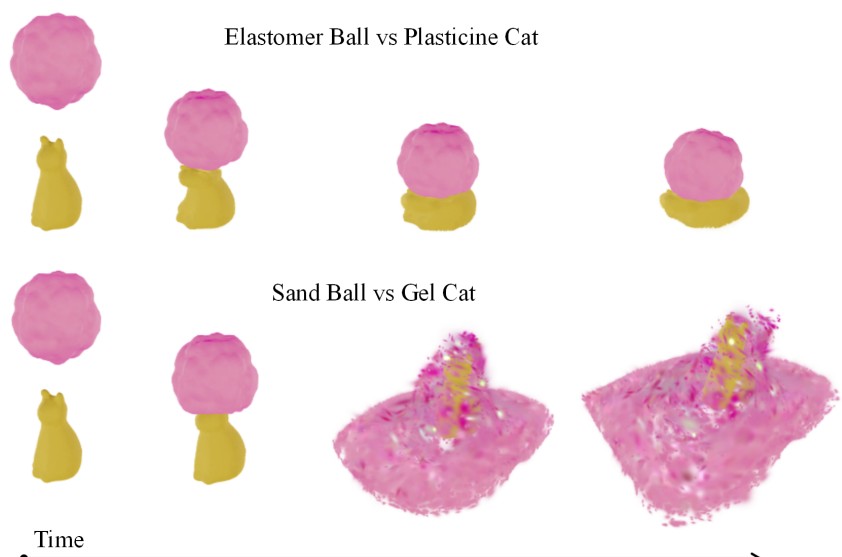

Elastomer Ball vs Plasticine Cat

Sand Ball vs Gel Cat

Time

Figure 8: **Multi-objects interaction with different materials properties.**

# E    ABLATIVE STUDY

This section presents ablation studies validating the effectiveness of our proposed modules, with quantitative results presented in Tab. 5. The tabulated results demonstrate that our EDCA framework and the Multi-Hypothesis Physics Verifier collectively yield significant improvements in optimization performance.

Table 5: **Ablative study in the synthesized dataset in Chamfer distance.**

| Material | Elastomer | Gel | Rubber | Plasticine | Granular | Non-Newtonian |
| Object | Ball | Duck | Pawn | Cat | Fish | Bottle |
|---|---|---|---|---|---|---|
| Ours w/o EDCA | 3.842 | 3.045 | 1.975 | 1.821 | 0.559 | 1.531 |
| Ours w/o Physics Verifiers | 1.024 | **0.694** | 0.243 | 0.510 | 0.084 | 0.769 |
| Ours | **0.922** | 0.702 | **0.200** | **0.318** | **0.077** | **0.757** |

# F    COMPUTATIONAL COST ANALYSIS

To provide a comprehensive analysis of the computational requirements of our method, we benchmarked PHYCO against the baseline NeuMA on the real-world dataset. All experiments were conducted on a single NVIDIA A800 80GB GPU to ensure a fair comparison.

The detailed results are presented in Table 6. As shown, our method demonstrates superior computational efficiency during the training phase. On average, PHYCO achieves its best performance in approximately **51.2 minutes**, which is significantly faster than NeuMA's average of **73.3 minutes**. The inference times for both methods are comparable, with our method being marginally faster. The memory footprint of our method is slightly higher, which is expected due to the additional components of the Edge-Aware Depth Consensus Anchors and the Multi-Hypothesis Physics Verifier. However, the increase is minimal and does not pose a significant overhead.

Table 6: Comparison of computational cost on the real-world dataset. In each cell, the format is: NeuMA / Ours. Best results are in **bold**.

| | Bun | Burger | Dog | Pig | Average |
|---|---|---|---|---|---|
| Training time (min) | **58.4** / 59.3 | 77.6 / **43.1** | 89.5 / **48.3** | 66.0 / **55.1** | 73.3 / **51.2** |
| Inference time (sec) | **31.9** / 32.0 | 32.7 / **31.1** | 32.5 / **31.5** | 32.3 / **31.2** | 32.4 / **31.5** |
| Memory Cost (GB) | **27.0** / 28.6 | **37.7** / 39.3 | **21.8** / 23.3 | **29.0** / 30.6 | **28.9** / 30.5 |

# G MATERIAL POINT METHOD FOR PHYSICAL SIMULATION

This section provides a systematic derivation of the Material Point Method (MPM) Jiang et al. (2016); Sulsky et al. (1995) time integration scheme (Algorithm 1) from continuum mechanics principles.

## G.1 GOVERNING EQUATIONS

The formulation begins with the Eulerian conservation laws. Mass conservation and momentum balance are expressed as:

$$\frac{D\rho}{Dt} = -\rho \nabla \cdot \mathbf{v}, \tag{10}$$

$$\rho \frac{D\mathbf{v}}{Dt} = \nabla \cdot \boldsymbol{\sigma} + \rho \mathbf{b}, \tag{11}$$

where $\rho$ is density, $D(\cdot)/Dt$ denotes material derivative, $\boldsymbol{\sigma}$ is Cauchy stress, and $\mathbf{b}$ represents body forces. Mass conservation is inherently enforced through Lagrangian particle advection.

## G.2 WEAK FORMULATION

The weak form of momentum balance is obtained by multiplying by a test function $\mathbf{w}$ and integrating over domain $\Omega$:

$$\int_\Omega \rho \mathbf{a} \cdot \mathbf{w} \, d\Omega = \int_\Omega (\nabla \cdot \boldsymbol{\sigma}) \cdot \mathbf{w} \, d\Omega + \int_\Omega \rho \mathbf{b} \cdot \mathbf{w} \, d\Omega. \tag{12}$$

Applying the divergence theorem yields:

$$\int_\Omega \rho \mathbf{a} \cdot \mathbf{w} \, d\Omega = -\int_\Omega \boldsymbol{\sigma} : \nabla \mathbf{w} \, d\Omega + \int_{\partial\Omega} \mathbf{w} \cdot \mathbf{T} \, dS \\ + \int_\Omega \rho \mathbf{b} \cdot \mathbf{w} \, d\Omega. \tag{13}$$

## G.3 SPATIAL DISCRETIZATION

MPM employs dual discretization with material points and background grid, leading to:

$$\sum_{b=1}^{G} M_{ab} \mathbf{a}_b = -\sum_{i=1}^{Q} V_i^0 \boldsymbol{\tau}_i \nabla N_a(\mathbf{x}_i) \\ + \sum_{i=1}^{Q} M_i N_a(\mathbf{x}_i) \mathbf{b}_i, \tag{14}$$

where:

- $M_{ab} = \sum_i M_i N_a(\mathbf{x}_i) N_b(\mathbf{x}_i)$ is consistent mass matrix
- $V_i^0$, $M_i$ are initial volume and mass
- $\boldsymbol{\tau}_i = J_i \boldsymbol{\sigma}_i$ denotes Kirchhoff stress
- $N_a(\cdot)$: grid basis function for node $a$
- $Q$, $G$: material point and grid node counts

## G.4 TEMPORAL DISCRETIZATION

Explicit Euler time integration gives:

$$\sum_{b=1}^{G} M_{ab} \frac{\mathbf{v}_b^{n+1} - \mathbf{v}_b^n}{\Delta t} = -\sum_{i=1}^{Q} V_i^0 \boldsymbol{\tau}_i^n \nabla N_a(\mathbf{x}_i^n) \\ + \sum_{i=1}^{Q} M_i N_a(\mathbf{x}_i^n) \mathbf{b}_i^n. \tag{15}$$

---

**Algorithm 3** MPM Algorithm

---

**Require:** Position $\mathbf{x}_i^n$, velocity $\mathbf{v}_i^n$, and elastic deformation gradient $\mathbf{F}_{e,i}^n$ for each material point $i = 1, \ldots, Q$ at time $t^n$.

**Ensure:** Updated position $\mathbf{x}_i^{n+1}$, velocity $\mathbf{v}_i^{n+1}$, and trial elastic deformation gradient $\mathbf{F}_{e,\text{trial},i}^{n+1}$ for each material point at time $t^{n+1}$.

1: **Particle-to-Grid Transfer:** For each grid node $b = 1, \ldots, G$, compute:

$$m_b^n = \sum_{i=1}^{Q} N_b(\mathbf{x}_i^n) \, M_i,$$

$$m_b^n \mathbf{v}_b^n = \sum_{i=1}^{Q} N_b(\mathbf{x}_i^n) \, M_i \, \mathbf{v}_i^n,$$

$$\mathbf{f}_{\sigma,b}^n = -\sum_{i=1}^{Q} J(\mathbf{F}_{e,i}^n) \, \frac{\rho_0}{M_i} \, \sigma(\mathbf{F}_{e,i}^n) \, \nabla N_b(\mathbf{x}_i^n),$$

$$\mathbf{f}_b^n = \sum_{i=1}^{Q} J(\mathbf{F}_{e,i}^n) \, \frac{\rho_0}{M_i} \, \mathbf{b}(\mathbf{x}_i^n) \, N_b(\mathbf{x}_i^n).$$

2: **Solve Eulerian Governing Equations:** For each grid node $b = 1, \ldots, G$, compute:

$$\dot{\mathbf{v}}_b^{n+1} = \frac{1}{m_b^n} \left( \mathbf{f}_{\sigma,b}^n + \mathbf{f}_b^n \right),$$

$$\Delta \mathbf{v}_b^{n+1} = \dot{\mathbf{v}}_b^{n+1} \Delta t,$$

$$\mathbf{v}_b^{n+1} = \mathbf{v}_b^n + \Delta \mathbf{v}_b^{n+1}.$$

3: **Grid-to-Particle Transfer:** For each material point $i = 1, \ldots, Q$, update:

$$\mathbf{v}_i^{n+1} = \sum_{b=1}^{G} N_b(\mathbf{x}_i^n) \, \mathbf{v}_b^{n+1},$$

$$\mathbf{F}_{e,\text{trial},i}^{n+1} = \left( \mathbf{I} + \Delta t \sum_{b=1}^{G} \mathbf{v}_b^{n+1} \otimes \nabla N_b(\mathbf{x}_i^n) \right) \mathbf{F}_{e,i}^n.$$

4: **Update Particle Positions:** For each material point $i = 1, \ldots, Q$, update:

$$\mathbf{x}_i^{n+1} = \mathbf{x}_i^n + \Delta t \, \mathbf{v}_i^{n+1}.$$

---

### G.5 ALGORITHMIC IMPLEMENTATION

The discretized system is implemented as Alg. 3 under MLS-MPM framework:

The neural constitutive models contribute to two critical components: 1) stress computation via neural elasticity law, and 2) plasticity correction through trial deformation gradient projection.

## H CONSTITUTIVE HYPOTHESIS

This section details our constitutive hypotheses for material modeling Ma et al. (2023), establishing four distinct constitutive assumptions for both elastic and plastic behaviors respectively.

### H.1 ELASTICITY MODELS

**1. Corotated Elasticity**

$$\mathbf{P} = 2\mu(\mathbf{F} - \mathbf{R})\mathbf{F}^\top + \lambda J(J-1)\mathbf{I} \tag{16}$$

- $\mathbf{F}$: Deformation gradient (input)
- $\mathbf{R} = \mathbf{U}\mathbf{V}^\top$: Rotation from SVD $\mathbf{F} = \mathbf{U}\boldsymbol{\Sigma}\mathbf{V}^\top$
- $J = \det(\mathbf{F})$: Volume change ratio
- $\mu = \frac{E}{2(1+\nu)}, \lambda = \frac{E\nu}{(1+\nu)(1-2\nu)}$: Lamé parameters

**2. St.Venant-Kirchhoff (StVK)**

$$\mathbf{P} = 2\mu\mathbf{F}\mathbf{E} + \lambda J(J-1)\mathbf{I}, \quad \mathbf{E} = \frac{1}{2}(\mathbf{F}^\top\mathbf{F} - \mathbf{I}) \tag{17}$$

- $\mathbf{E}$: Green-Lagrange strain tensor
- Maintains same $\mu, \lambda$ definition as Corotated

**3. Volume Elasticity** Mode-dependent pressure term:

$$\mathbf{P} = \begin{cases} \kappa(J - J^{-\gamma+1})\mathbf{I} & \text{(Ziran)} \\ \lambda J(J-1)\mathbf{I} & \text{(Taichi)} \end{cases} \tag{18}$$

- $\kappa = \frac{2}{3}\mu + \lambda$: Bulk modulus
- $\gamma$: Adiabatic index (default 2)

**4. Sigma Elasticity** Logarithmic strain formulation:

$$\mathbf{P} = \mathbf{U}[\mathrm{diag}(2\mu\ln\sigma_i + \lambda\sum\ln\sigma_j)]\mathbf{U}^\top \tag{19}$$

- $\sigma_i$: Singular values of $\mathbf{F}$
- Strain defined as $\epsilon_i = \ln\sigma_i$

### H.2 PLASTICITY MODELS

**1. Identity Plasticity**

$$\mathbf{F}^p = \mathbf{F} \tag{20}$$

- No plasticity effect

**2. Sigma Plasticity** Volumetric preservation:

$$\mathbf{F}^p = J^{1/3}\mathbf{I} \tag{21}$$

- Enforces $J = \det(\mathbf{F}^p) = 1$

**3. Von Mises Plasticity** Yield condition and strain update:

$$\|\epsilon_{\mathrm{dev}}\| \geq \frac{\sigma_y}{2\mu}, \quad \epsilon \leftarrow \epsilon - \Delta\gamma\frac{\epsilon_{\mathrm{dev}}}{\|\epsilon_{\mathrm{dev}}\|} \tag{22}$$

- $\epsilon_{\mathrm{dev}} = \epsilon - \frac{1}{3}\mathrm{tr}(\epsilon)\mathbf{I}$: Deviatoric strain
- $\sigma_y$: Yield stress

**4. Drucker-Prager Plasticity** Frictional yield criterion:

$$\alpha\mathrm{tr}(\epsilon) + \|\epsilon_{\mathrm{dev}}\| \geq c \tag{23}$$

- $\alpha = \frac{2\sqrt{2}\sin\phi}{3-\sin\phi}$: Friction parameter
- $c$: Cohesion, $\phi$: Friction angle

## I PREPROCESSING GAUSSIAN KERNELS FOR SIMULATION

A fundamental challenge in applying physics to scenes reconstructed via 3D Gaussian Splatting is that the representation is superficial; the Gaussians are concentrated on the object's exterior, creating a hollow shell. Such models fail to exhibit realistic volumetric dynamics, often collapsing under external forces. To overcome this limitation, we propose a procedure to densify the interior volume.

Our method populates the void regions by first interpreting the collection of surface Gaussians as a continuous opacity field. This field is then rasterized onto a 3D volumetric grid. We employ a robust ray-casting technique to classify grid cells as either internal or external. A cell is designated as internal if probes sent out in multiple directions all intersect regions of high opacity, confirming it is enclosed by the object's surface. To enhance accuracy, we verify this condition by checking the number of surface crossings.

Each particle seeded in the interior must be initialized with appropriate attributes. We assign visual properties, such as opacity and spherical harmonics, by sampling from the closest particle in the original surface reconstruction. The covariance matrix for each new particle is initialized as an isotropic sphere, with a radius computed from its representative volume $V_O^P$. This densification ensures that the simulated object has a proper internal structure, allowing for the accurate simulation of volumetric effects and preventing unrealistic structural failures.

## J THEORETICAL ANALYSIS

In this section, we provide a theoretical analysis of the convergence properties of our proposed optimization algorithm. Our goal is to prove that the optimization of the total loss function, regularized by our Multi-Hypothesis Physics Verifier, converges to a stationary point.

### ASSUMPTIONS

To facilitate the proof, we make the following reasonable assumptions.

**Assumption 1** (Structure of the Ground Truth Model). *We assume that the true constitutive laws for elasticity, $\mathcal{E}^*$, and plasticity, $\mathcal{P}^*$, can be decomposed into a dominant, explicit model from our hypothesis sets $(\mathcal{H}_e, \mathcal{H}_p)$ plus a minor perturbation term $(\delta_e, \delta_p)$.*

$$\mathcal{E}^* = \mathcal{H}_e^j + \delta_e \quad and \quad \mathcal{P}^* = \mathcal{H}_p^k + \delta_p, \tag{24}$$

*where $\mathcal{H}_e^j \in \mathcal{H}_e$ and $\mathcal{H}_p^k \in \mathcal{H}_p$ are the ground truth explicit models. The perturbation terms are assumed to be small, i.e., their norms are bounded: $||\delta_e|| \leq \epsilon_\delta$ and $||\delta_p|| \leq \epsilon_\delta$ for some small $\epsilon_\delta > 0$.*

**Assumption 2** (Expressiveness of the Neural Network). *We assume that the neural networks $\mathcal{E}_{\theta_e}$ and $\mathcal{P}_{\theta_p}$ are universal approximators, possessing sufficient capacity to represent the true constitutive laws. This implies the existence of optimal parameters $\theta_e^*$ and $\theta_p^*$ such that $\mathcal{E}_{\theta_e^*} = \mathcal{E}^*$ and $\mathcal{P}_{\theta_p^*} = \mathcal{P}^*$.*

**Assumption 3** (Smoothness and Boundedness). *The total loss function $L(\theta)$, the neural network models $\mathcal{E}_{\theta_e}$ and $\mathcal{P}_{\theta_p}$, and all explicit hypotheses $\mathcal{H}$ are Lipschitz continuous with respect to their inputs and parameters. This implies that their gradients are bounded.*

**Assumption 4** (Well-posedness of the Inverse Problem). *The inverse problem of solving for the physical parameters $\Theta$ in Algorithm 2 (the 'argmin' step) is locally well-posed. When the neural model's output is close to that of an explicit model, the estimated parameters are unique and stable.*

### ANALYSIS OF THE PHYSICS VERIFIER $\mathcal{R}$

We first prove a key lemma regarding the behavior of our physics-based regularizer, $\mathcal{R}$.

**Lemma 1** (Properties of the Physics Verifier). *Under Assumptions 1-4, when the neural network model $\mathcal{E}_{\theta_e}$ is sufficiently close to the ground truth model $\mathcal{E}^*$, the physics verifier $\mathcal{R}_e$ provides a meaningful penalty that is minimized as $\mathcal{E}_{\theta_e} \to \mathcal{E}^*$. Its gradient guides the optimization towards the structure dominated by the true explicit model $\mathcal{H}_e^j$.*

*Proof.* Let the error between the current network and the true model be $\Delta_e$, such that $\mathcal{E}_{\theta_e} = \mathcal{E}^* + \Delta_e = (\mathcal{H}_e^j + \delta_e) + \Delta_e$.

When the verifier evaluates the correct hypothesis $\mathcal{H}_e^j$, it attempts to fit $\mathcal{H}_e^j(\Theta_e^j)$ to the output of $\mathcal{E}_{\theta_e}$. Since the dominant component of $\mathcal{E}_{\theta_e}$ is $\mathcal{H}_e^j$, by Assumption 4, the estimated parameters $\hat{\Theta}_e^j$ will be stable across different material points. Consequently, the variance $var\{\hat{\Theta}_e^j\}$ will be small, and the corresponding credibility weight $\omega_e^j$ will be large.

Conversely, for any incorrect hypothesis $\mathcal{H}_e^l$ where $l \neq j$, fitting it to the data generated by $\mathcal{E}_{\theta_e}$ will result in unstable parameter estimates $\hat{\Theta}_e^l$ with high variance. Thus, the weight $\omega_e^l$ will be close to zero.

As a result, the summation for the physics residual $\mathcal{R}_e^t$ will be dominated by the term corresponding to the true hypothesis $\mathcal{H}_e^j$:

$$\mathcal{R}_e^t = \sum_{k=1}^{K} \omega_e^k ||\mathcal{E}_{\theta_e} - \mathcal{H}_e^k(\cdot; \mathbb{E}[\hat{\Theta}_e^k])||_F^2 \approx \omega_e^j ||\mathcal{E}_{\theta_e} - \mathcal{H}_e^j(\cdot; \mathbb{E}[\hat{\Theta}_e^j])||_F^2. \tag{25}$$

Substituting the expression for $\mathcal{E}_{\theta_e}$ and noting that $\mathbb{E}[\hat{\Theta}_e^j]$ approximates the true parameters of $\mathcal{H}_e^j$, we get:

$$\mathcal{R}_e^t \approx \omega_e^j ||(\mathcal{H}_e^j + \delta_e + \Delta_e) - \mathcal{H}_e^j||_F^2 = \omega_e^j ||\delta_e + \Delta_e||_F^2. \tag{26}$$

This shows that the verifier penalizes the deviation $\Delta_e$ of the neural network from the true model structure. Minimizing $\mathcal{R}_e^t$ with gradient descent therefore corresponds to minimizing $||\Delta_e||_F^2$, pushing $\mathcal{E}_{\theta_e}$ towards $\mathcal{E}^*$. A symmetric argument holds for the plasticity model $\mathcal{P}_{\theta_p}$. $\qquad\square$

PROOF OF CONVERGENCE

With the behavior of the regularizer established, we can now prove the convergence of the overall algorithm.

**Theorem 1** (Convergence to a Stationary Point). *Under Assumptions 1-4, the optimization of the total loss function $L(\theta) = \lambda_g \mathcal{L}_{geo} + \lambda_m \mathcal{L}_{mask} + \mathcal{R}$ via gradient descent with a sufficiently small learning rate $\eta$ ensures that the gradient of the loss function converges to zero:*

$$\lim_{k \to \infty} ||\nabla L(\theta_k)|| = 0. \tag{27}$$

*Proof.* Let $L(\theta)$ be the total loss function. As the component losses ($\mathcal{L}_{geo}$, $\mathcal{L}_{mask}$) and the regularizer $\mathcal{R}$ are non-negative, the loss function $L(\theta)$ is bounded below by 0.

The gradient descent update rule is $\theta_{k+1} = \theta_k - \eta \nabla L(\theta_k)$. From Assumption 3 (Lipschitz continuity), the Descent Lemma states that for a sufficiently small learning rate $\eta > 0$ (specifically, $\eta < 2/L_{smooth}$ where $L_{smooth}$ is the Lipschitz constant of $\nabla L$), the loss decreases at each step unless the gradient is zero:

$$L(\theta_{k+1}) \leq L(\theta_k) - \frac{\eta}{2} ||\nabla L(\theta_k)||^2. \tag{28}$$

This inequality shows that the sequence of loss values $\{L(\theta_k)\}$ is monotonically decreasing. Since it is also bounded below, the Monotone Convergence Theorem guarantees that the sequence converges to a finite limit $L^*$.

Summing the inequality from $k = 0$ to $N$:

$$\sum_{k=0}^{N} (L(\theta_k) - L(\theta_{k+1})) \geq \frac{\eta}{2} \sum_{k=0}^{N} ||\nabla L(\theta_k)||^2. \tag{29}$$

The left-hand side is a telescoping sum, which simplifies to $L(\theta_0) - L(\theta_{N+1})$. As $N \to \infty$, this converges to the finite value $L(\theta_0) - L^*$.

$$L(\theta_0) - L^* \geq \frac{\eta}{2} \sum_{k=0}^{\infty} ||\nabla L(\theta_k)||^2. \tag{30}$$

Since the sum of the series $\sum ||\nabla L(\theta_k)||^2$ is finite, its terms must converge to zero. Therefore, we conclude that $\lim_{k \to \infty} ||\nabla L(\theta_k)||^2 = 0$, which implies that the norm of the gradient itself converges to zero. $\qquad\square$

## K LLM USAGE STATEMENT

The authors employed Google Gemini 2.5 Pro to assist in the writing process of this manuscript. Specifically, the model was used for rephrasing sentences to improve clarity, structuring paragraphs for better flow, and polishing the overall language of the paper. The core scientific ideas, experimental results, and their interpretation were solely conceived by the human authors, who are fully responsible for all content presented.

