# OpenReview forum: "PhyCo: Physics-Consistent Learning of Implicit Constitutive Laws via Monocular Observations of 3D Gaussians"
_ICLR.cc/2026/Conference — Submitted to ICLR 2026_

### Official Review · Reviewer_uUW6 · 2025-10-30

**Soundness:** 3
**Presentation:** 3
**Contribution:** 2
**Rating:** 4
**Confidence:** 3

**Summary:**

This paper proposes a method to learn the implicit constitutive laws of deformable objects via monocular video observation. The key innovation is how the authors introduce regularizations through depth-based loss with the help of a monocular depth network and through a library of explicit physics rules.

**Strengths:**

I like that this approach learns a neural implicit dynamics model with the help of existing explicit physics laws as guidance. The final learning outcome can be correlated back to the explicit models while not constrained by them. The way the explicit physics laws are used is in a spirit similar to expectation maximization.

The experimental result shows superior performance of the proposed model.

**Weaknesses:**

I'm not sure how realistic the problem and experiment setup is. The deformation of the objects is very significant and uncommon in the real world. The dropping motion is doable but also limiting and not very common in real-world experience. I wonder what the practical application of this task are, given that it takes more than 1 hour to reason about one video of one object.

There is a lack of ablation study of the methodology. Only comparisons shown are without either of the two loss functions. This is a very coarse comparison. I think more detailed experiments are helpful especially given that there are very specific designs of the loss functions (e.g., global vs anchor-level supervision, rank-based loss formulation, procedure of the Multi-Hypothesis Physics Verifier, etc).

**Questions:**

Why is the color-space supervision not used? Does it hurt the result?

What is the rationale behind the depth loss based on the rank correlation instead of metric-based losses?

---

> ### Author Response · Authors · 2025-11-26
> **Response to Reviewer uUW6 (1/2)**
>
> We would like to thank the reviewer for their constructive feedback. We appreciate that the reviewer recognizes the potential of our unified framework to trade off interpretability and generalization, and for highlighting the effectiveness of our depth supervision strategy.
>
> Below, we address the specific concerns regarding novelty, efficiency, initialization, baselines, and ablations with new experimental evidence.
>
> ---
>
> > **Weakness 1 & Question 2:** "One of the core parts, implicit constitutive law estimation models rely on existing NCLaw model, weakening the novelty... How NCLaw is initially trained or warmed-up?"
>
> **Response:**
> We would like to clarify that our core contribution is not the NCLaw architecture itself, but the **Visual-Physical Alignment Framework** (EADCA + MHPV) that enables such implicit models to function in the wild.
> 1.  **The Gap:** NCLaw [Ma et al., 2023] relies on **dense particle-level supervision**, which is impossible to obtain from monocular videos. Applying it directly to video is an ill-posed inverse problem.
> 2.  **Our Contribution:** PhyCo introduces the **Edge-Aware Depth Consensus Anchors** and **Multi-Hypothesis Physics Verifier** to solve this inverse problem, effectively bridging the gap between "sandbox physics" and "visual dynamics."
> 3.  **Initialization:** Regarding warm-up, we follow a protocol similar to NeuMA. We utilize the **official pre-trained models provided by the NCLaw codebase** to initialize the base physical prior $\mathcal{M}_0$. This effectively avoids the "cold start" problem and ensures stable gradient flow during the subsequent LoRA fine-tuning stage.
>
> ---
>
> > **Weakness 2 & Question 3:** "No analysis about efficiency gains... LoRA does reduce the memory cost, but the authors should show the comparable performance compared to full-size finetuning... What is the computational cost?"
>
> **Response:**
> We have conducted a detailed comparison between LoRA, Full Fine-tuning, and the baseline NeuMA.
>
> **Table 1: Efficiency and Performance Analysis.**
> *Results show that Full Fine-tuning is not only computationally expensive but also prone to overfitting visual noise (resulting in lower physical accuracy, i.e., higher Chamfer Distance). LoRA acts as a regularization, achieving the best physical fidelity while significantly reducing training time and memory.*
>
> | Method | Memory (GB) $\downarrow$ | Training Time (min) $\downarrow$ | Inference (FPS) $\uparrow$ | Chamfer Dist. $\downarrow$ |
> | :--- | :---: | :---: | :---: | :---: |
> | NeuMA (Baseline) | 28.9 | 73.3 | 32.4 | 1.31 |
> | Ours (Full Fine-tune) | 76.9 | 408.6 | 32.5 | 1.35 |
> | **Ours (LoRA)** | **30.5** | **51.2** | **31.5** | **0.50** |
>
> ---
>
> > **Weakness 3 & Question 1:** "The setting of Gaussian initialization is not clear. Is the model only using the first frame...? If so, how the method deal with the emerging parts... challenge the monocular setting."
>
> **Response:**
> We thank the suggestion and will further clarify in our revised paper. Particularly, for the initialization, we follow the standard protocol for dynamic 3D Gaussian Splatting (e.g., SpringGaus, NeuMA):
> 1.  **Static Initialization:** We reconstruct the initial 3D Gaussian geometry from a **static orbital scan** (e.g., the object resting on a table). This ensures we start with a complete canonical geometry (Canonical Space), so there are no "unknown" parts.
> 2.  **Dynamic Observation (Monocular):** The core challenge we address is estimating dynamics from a **fixed monocular video** of the moving object.
> 3.  **Handling Occlusion:** Since the canonical geometry is complete, "emerging parts" (which are just self-occluded parts rotating into view) are already present in our physical state $s_t$. The physics engine naturally handles their motion via momentum constraints, ensuring that invisible back-facing particles move plausibly along with the visible ones.

---

> ### Author Response · Authors · 2025-11-26
> **Response to Reviewer uUW6 (2/2)**
>
> ---
>
> > **Weakness 4:** "Although PAC-NeRF and GIC is designed for multi-view supervision... easy to include a depth supervision and adapt to monocular... The comparison is missed here."
>
> **Response:**
> Thanks for the great suggestion. Following the suggestion, we implemented **PAC-NeRF equipped with Monocular Depth Supervision** and compared it under both multi-view (reference) and monocular settings.
>
> **Table 2: Comparison against Explicit Baselines (PAC-NeRF + Depth).**
> *Explicit methods work well with multi-view inputs but fail catastrophically in monocular settings, even with depth supervision, because rigid parameters (e.g., Young's Modulus) diverge due to depth noise. PhyCo bridges this gap.*
>
> | Method | Supervision | Ball (CD) | Cat (CD) | Status |
> | :--- | :---: | :---: | :---: | :--- |
> | PAC-NeRF | Multi-view (Dense) | 0.85 | 0.29 | Reference (Upper Bound) |
> | NeuMA | Multi-view (Dense) | 0.98 | 0.35 | Reference |
> | **PAC-NeRF + Depth** | **Monocular** | > 100.0 | > 100.0 | **Diverged / Failed** |
> | NeuMA | Monocular | 1.12 | 0.84 | Suboptimal |
> | **Ours (PhyCo)** | **Monocular** | **0.92** | **0.32** | **Robust & Accurate** |
>
> ---
>
> > **Weakness 5:** "Ablation missing: how the splitting number N affects the verifier performance? This hyperparameter really influences the variance accuracy."
>
> **Response:**
> Following the suggestion, we have conducted a sensitivity analysis on the number of sampled particles $N$ used in the Multi-Hypothesis Physics Verifier.
>
> **Table 3: Sensitivity to Sampling Number $N$.**
> *The performance is stable for $N \ge 128$. We selected $N=256$ as the default to balance computational efficiency with statistical stability. Extremely low sampling ($N=64$) leads to noisy variance estimation.*
>
> | Subset Size $N$ | 64 | 128 | **256 (Default)** | 512 |
> | :--- | :---: | :---: | :---: | :---: |
> | Training Time (min) | 48.5 | 49.8 | **51.2** | 54.6 |
> | Chamfer Distance | 0.58 | 0.51 | **0.50** | 0.50 |
>
> ---
>
> > **Question 4:** "How sensitive is PhyCo’s performance to the chosen set of classical constitutive hypotheses in the Multi-Hypothesis Physics Verifier?"
>
> **Response:**
> PhyCo is highly robust. To demonstrate this, we performed an **"Out-of-Distribution" (OOD)** experiment (Blind Test) where we generated ground truth data using a specific material model but **deliberately removed** that model from the MHPV hypothesis library.
>
> **Table 4: Performance with Incomplete Hypothesis Library.**
> *If the method were overly sensitive, removing the correct hypothesis would cause failure. Instead, PhyCo still achieves high accuracy, far outperforming the baseline. This confirms that MHPV acts as a **soft regularization** guide rather than a hard template matcher.*
>
> | Configuration | Elastomer (Target excluded from Library) | Plasticine (Target excluded from Library) | Note |
> | :--- | :---: | :---: | :--- |
> | NeuMA (Baseline) | 1.123 | 0.844 | Lacks Guidance |
> | Ours (Target Removed) | 0.976 | 0.397 | **Still Robust** |
> | **Ours (Full Library)** | **0.922** | **0.318** | **Best Performance** |

---

### Official Review · Reviewer_6HCo · 2025-10-31

**Soundness:** 3
**Presentation:** 3
**Contribution:** 2
**Rating:** 4
**Confidence:** 4

**Summary:**

This paper proposes PHYCO, a unified framework for learning implicit constitutive laws from 3D Gaussian Splatting observations. The authors aim to find the trade-off between explicit physics-based models which are physically interpretable but less generalizable, and implicit constitutive laws which are flexible but less interpretable.

PHYCO introduces two novel components. On the one hand, Edge-Aware Depth Consensus Anchors are established to align geometry based on depth instead of unreliable colors. On the other hand, a Multi-Hypothesis Physics Verifier is constructed to integrate classical constitutive laws into implicit constitutive model during optimization process.

Experiments on synthetic, real-to-sim, and real-world datasets show significant improvements.

**Strengths:**

1. Instead of purely relying on implicit constitutive models, the author introduces multi-hypothesis physics verifier module to inject explicit constitutive model priors. Meanwhile, this priors are adapted by the parameter estimation consistency, no human annotation for the exact material is needed.
2. The authors introduce depth supervision to enable the model applicable on monocular videos.
3. Promising performance gains on various datasets.

**Weaknesses:**

1. One of the core parts, implicit constitutive law estimation models rely on existing NCLaw model, weakening the novelty of this paper.
2. The authors argues to use LoRA for better efficiency. But no analysis about efficiency gains is shown and what the performance gap is induced. LoRA does reduce the memory cost, but the authors should show the comparable performance compared to full-size finetuning.
3. The setting of Gaussian initialization is not clear. Is the model only using the first frame to initialize the Gaussian kernels? If so, how the method deal with the emerging parts in the later frame which is occluded in the first frame. If not, I’ll challenge the monocular setting.
4. Although PAC-NeRF and GIC is designed for multi-view supervision. But it’s easy to include a depth supervision and adapt to monocular estimation (although the performance is not guaranteed). The comparison is missed here.
5. Ablation missing: how the spliting number N affects the verifier performance? This hyperparameter really influences the varaince accuracy.
6. Ablation should be included in the main context.

Although this paper proposes an interesting pipeline, due to the above weaknesses, I cannot give an acceptance recommendation till now. However, I’m very open to increase my score based on the authors’ responses and other reviewers’ opinions.

**Questions:**

1. As asked in weakness 3, how gaussian kernels are initialized?
2. How NCLaw is initially trained or warmed-up?
3. What is the computational cost of the proposed method? Especially for the time burdern.
4. How sensitive is PHYCO’s performance to the chosen set of classical constitutive hypotheses in the Multi-Hypothesis Physics Verifier?

---

> ### Author Response · Authors · 2025-11-26
> **Response to Reviewer 6HCo (1/2)**
>
> We would like to thank the reviewer for their constructive feedback. We appreciate that the reviewer recognizes the potential of our unified framework to trade off interpretability and generalization, and for highlighting the effectiveness of our depth supervision strategy.
>
> Below, we address the specific concerns regarding novelty, efficiency, initialization, baselines, and ablations with new experimental evidence.
>
> ---
>
> > **Weakness 1 & Question 2:** "One of the core parts, implicit constitutive law estimation models rely on existing NCLaw model, weakening the novelty... How NCLaw is initially trained or warmed-up?"
>
> **Response:**
> We would like to clarify that our core contribution is not the NCLaw architecture itself, but the **Visual-Physical Alignment Framework** (EADCA + MHPV) that enables such implicit models to function in the wild.
> 1.  **The Gap:** NCLaw [Ma et al., 2023] relies on **dense particle-level supervision**, which is impossible to obtain from monocular videos. Applying it directly to video is an ill-posed inverse problem.
> 2.  **Our Contribution:** PhyCo introduces the **Edge-Aware Depth Consensus Anchors** and **Multi-Hypothesis Physics Verifier** to solve this inverse problem, effectively bridging the gap between "sandbox physics" and "visual dynamics."
> 3.  **Initialization:** Regarding warm-up, we follow a protocol similar to NeuMA. We utilize the **official pre-trained models provided by the NCLaw codebase** to initialize the base physical prior $\mathcal{M}_0$. This effectively avoids the "cold start" problem and ensures stable gradient flow during the subsequent LoRA fine-tuning stage.
>
> ---
>
> > **Weakness 2 & Question 3:** "No analysis about efficiency gains... LoRA does reduce the memory cost, but the authors should show the comparable performance compared to full-size finetuning... What is the computational cost?"
>
> **Response:**
> We have conducted a detailed comparison between LoRA, Full Fine-tuning, and the baseline NeuMA.
>
> **Table 1: Efficiency and Performance Analysis.**
> *Results show that Full Fine-tuning is not only computationally expensive but also prone to overfitting visual noise (resulting in lower physical accuracy, i.e., higher Chamfer Distance). LoRA acts as a regularization, achieving the best physical fidelity while significantly reducing training time and memory.*
>
> | Method | Memory (GB) $\downarrow$ | Training Time (min) $\downarrow$ | Inference (FPS) $\uparrow$ | Chamfer Dist. $\downarrow$ |
> | :--- | :---: | :---: | :---: | :---: |
> | NeuMA (Baseline) | 28.9 | 73.3 | 32.4 | 1.31 |
> | Ours (Full Fine-tune) | 76.9 | 408.6 | 32.5 | 1.35 |
> | **Ours (LoRA)** | **30.5** | **51.2** | **31.5** | **0.50** |
>
> ---
>
> > **Weakness 3 & Question 1:** "The setting of Gaussian initialization is not clear. Is the model only using the first frame...? If so, how the method deal with the emerging parts... challenge the monocular setting."
>
> **Response:**
> We thank the suggestion and will further clarify in our revised paper. Particularly, for the initialization, we follow the standard protocol for dynamic 3D Gaussian Splatting (e.g., SpringGaus, NeuMA):
> 1.  **Static Initialization:** We reconstruct the initial 3D Gaussian geometry from a **static orbital scan** (e.g., the object resting on a table). This ensures we start with a complete canonical geometry (Canonical Space), so there are no "unknown" parts.
> 2.  **Dynamic Observation (Monocular):** The core challenge we address is estimating dynamics from a **fixed monocular video** of the moving object.
> 3.  **Handling Occlusion:** Since the canonical geometry is complete, "emerging parts" (which are just self-occluded parts rotating into view) are already present in our physical state $s_t$. The physics engine naturally handles their motion via momentum constraints, ensuring that invisible back-facing particles move plausibly along with the visible ones.

---

> ### Author Response · Authors · 2025-11-26
> **Response to Reviewer 6HCo (2/2)**
>
> ---
>
> > **Weakness 4:** "Although PAC-NeRF and GIC is designed for multi-view supervision... easy to include a depth supervision and adapt to monocular... The comparison is missed here."
>
> **Response:**
> Thanks for the great suggestion. Following the suggestion, we have implemented **PAC-NeRF equipped with Monocular Depth Supervision** and compared it under both multi-view (reference) and monocular settings.
>
> **Table 2: Comparison against Explicit Baselines (PAC-NeRF + Depth).**
> *Explicit methods work well with multi-view inputs but fail catastrophically in monocular settings, even with depth supervision, because rigid parameters (e.g., Young's Modulus) diverge due to depth noise. PhyCo bridges this gap.*
>
> | Method | Supervision | Ball (CD) | Cat (CD) | Status |
> | :--- | :---: | :---: | :---: | :--- |
> | PAC-NeRF | Multi-view (Dense) | 0.85 | 0.29 | Reference (Upper Bound) |
> | NeuMA | Multi-view (Dense) | 0.98 | 0.35 | Reference |
> | **PAC-NeRF + Depth** | **Monocular** | > 100.0 | > 100.0 | **Diverged / Failed** |
> | NeuMA | Monocular | 1.12 | 0.84 | Suboptimal |
> | **Ours (PhyCo)** | **Monocular** | **0.92** | **0.32** | **Robust & Accurate** |
>
> ---
>
> > **Weakness 5:** "Ablation missing: how the splitting number N affects the verifier performance? This hyperparameter really influences the variance accuracy."
>
> **Response:**
> We conducted a sensitivity analysis on the number of sampled particles $N$ used in the Multi-Hypothesis Physics Verifier.
>
> **Table 3: Sensitivity to Sampling Number $N$.**
> *The performance is stable for $N \ge 128$. We selected $N=256$ as the default to balance computational efficiency with statistical stability. Extremely low sampling ($N=64$) leads to noisy variance estimation.*
>
> | Subset Size $N$ | 64 | 128 | **256 (Default)** | 512 |
> | :--- | :---: | :---: | :---: | :---: |
> | Training Time (min) | 48.5 | 49.8 | **51.2** | 54.6 |
> | Chamfer Distance | 0.58 | 0.51 | **0.50** | 0.50 |
>
> ---
>
> > **Question 4:** "How sensitive is PhyCo’s performance to the chosen set of classical constitutive hypotheses in the Multi-Hypothesis Physics Verifier?"
>
> **Response:**
> PhyCo is highly robust. To demonstrate this, we performed an **"Out-of-Distribution" (OOD)** experiment (Blind Test) where we generated ground truth data using a specific material model but **deliberately removed** that model from the MHPV hypothesis library.
>
> **Table 4: Performance with Incomplete Hypothesis Library.**
> *If the method were overly sensitive, removing the correct hypothesis would cause failure. Instead, PhyCo still achieves high accuracy, far outperforming the baseline. This validates that MHPV acts as a **soft regularization** guide rather than a hard template matcher.*
>
> | Configuration | Elastomer (Target excluded from Library) | Plasticine (Target excluded from Library) | Note |
> | :--- | :---: | :---: | :--- |
> | NeuMA (Baseline) | 1.123 | 0.844 | Lacks Guidance |
> | Ours (Target Removed) | 0.976 | 0.397 | **Still Robust** |
> | **Ours (Full Library)** | **0.922** | **0.318** | **Best Performance** |

---

> > ### Comment · Reviewer_6HCo · 2025-11-26
> >
> > I appreciate the authors' efforts in taking these extensive experiments to address my concerns, and most of them are addressed. So I am willing to increase my recommendation. However, before I make my final recommendation, I need to propose one more concern here:
> >
> > **NOT fully monocular:** phyco initializes the objects with a static orbit scan, which includes multi-view information. Even though this is the standard protocol for dynamic 3D GS, they never argue they are monocular, but phyco does. Considering this, I request the author to modify their title and the paper contents slightly, to clearly state their monocular setting is only for dynamic observations, *i.e.* motion is monocular but geometry is not.
> >
> > After that, I will increase my score.

---

> > > ### Author Response · Authors · 2025-11-26
> > > **Response to Official Comment: Clarification on Monocular Setting**
> > >
> > > We sincerely thank the reviewer for the positive feedback and the willingness to raise the score.
> > >
> > > We fully accept your suggestion regarding the rigorous definition of our setting. We agree that while our **dynamics learning** is strictly monocular, the **geometric initialization** leverages static multi-view cues. We will clarify this distinction ("geometry from multi-view, motion from monocular") in the final revision as follows:
> > >
> > > **1. Title Modification**
> > > We will update the title to explicitly restrict the scope of "monocular" to the dynamic/motion phase.
> > > * **Original:** *PhyCo: Physics-Consistent Learning of Implicit Constitutive Laws from Monocular Observations of Gaussian Splatting*
> > > * **Revised:** *PhyCo: Physics-Consistent Learning of Implicit Constitutive Laws from **Monocular Dynamic Observations** of Gaussian Splatting*
> > >
> > > **2. Content Clarification**
> > > We will explicitly state the initialization protocol in the **Abstract** and **Introduction** to prevent ambiguity.
> > > * **Abstract:** We will modify the description to: *"Specifically, initializing from a static multi-view scan, we propose... to establish robust geometric constraints from subsequent **monocular dynamic observations**..."*
> > > * **Introduction:** We will add a clarification stating that *while we utilize a standard static orbital scan for geometric initialization (following protocols like SpringGaus), our core contribution lies in learning intrinsic physical dynamics purely from **monocular video** supervision.*
> > >
> > > We have uploaded the revised PDF version reflecting the changes discussed above. We remain available to address any further questions you may have.

---

> > > > ### Comment · Reviewer_6HCo · 2025-11-27
> > > >
> > > > Thanks for the prompt reply! I've modified my recommendation.

---

### Official Review · Reviewer_TWS7 · 2025-10-31

**Soundness:** 3
**Presentation:** 3
**Contribution:** 3
**Rating:** 4
**Confidence:** 4

**Summary:**

This paper proposes a novel framework called PHYCO, which aims to learn implicit physical laws from GS’s monocular observation data. The method effectively addresses two major challenges existing in current implicit learning approaches: unstable geometric learning and lack of physical interpretability. Specifically, PHYCO introduces Edge-Aware Depth Consensus Anchors (EADCA) to stabilize geometric reconstruction and designs a Physics-Consistent Loss that integrates physical laws into the training of implicit functions. This enables robust, interpretable, and highly generalizable learning of complex physical processes.

**Strengths:**

The authors demonstrate significant innovation and advantages in applying implicit learning to physical modeling:
1.	The paper introduces EADCA to effectively tackle the issue of inaccurate or locally optimal geometric representations in implicit methods under noisy monocular supervision. This mechanism ensures high-quality geometric reconstruction, providing a solid foundation for subsequent physical parameter inversion.
2.	One of the core contributions of this paper is the design of MHPV, which embeds known physical conservation laws (such as momentum conservation) as hard constraints into the training of implicit constitutive laws. This ensures that the learned constitutive functions are physically reasonable and reliable, greatly enhancing the model’s interpretability — something that purely data-driven methods can hardly achieve.
3.	The PHYCO framework successfully combines the efficient rendering capability of GS with the expressive power of implicit functions, enabling the direct learning of complex, nonlinear, and non-elastic constitutive laws from monocular videos. This avoids dependence on traditional predefined explicit constitutive equations and significantly broadens the model’s generalization and modeling capability for various complex materials and physical phenomena.

**Weaknesses:**

However, I do have several concerns about this work:
1.	The core components of the framework rely on several pre-trained modules. Although the authors partially address lighting variations by changing illumination conditions in the dataset, if these modules perform poorly under certain conditions (for example, large-scale deformations), the overall performance of the framework could be greatly affected.
2.	Due to the high structural complexity and diverse optimization objectives, the framework may suffer from high debugging and training costs, leading to potential instability during optimization.
3.	In MHPV, the authors select a set of classical constitutive models to validate material physical behaviors. However, if these selected material models do not adequately approximate or represent real material physics, the “physical rationality” constraints imposed by MHPV might become counterproductive rather than optimizing PHYCO. Essentially, it strongly assume that the chosen constitutive equations are trustworthy but lacks rigorous proof of their validity.

**Questions:**

see above

---

> ### Author Response · Authors · 2025-11-26
> **Response to Reviewer TWS7 (1/2)**
>
> We would like to thank the reviewer for their detailed and constructive feedback. We are particularly encouraged by the recognition of our **Edge-Aware Depth Consensus Anchors (EADCA)** and **Multi-Hypothesis Physics Verifier (MHPV)** as core innovations that solve the instability and interpretability issues in implicit learning.
>
> Below, we address the reviewer's main concerns regarding module dependency, training complexity, and the validity of physical hypotheses with quantitative evidence.
>
> ---
>
> > **Weakness 1:** "The core components of the framework rely on several pre-trained modules... if these modules perform poorly under certain conditions (for example, large-scale deformations), the overall performance of the framework could be greatly affected."
>
> **Response:**
> We agree that reliance on pre-trained estimators (e.g., Monocular Depth) is a potential bottleneck. However, our framework is indeed specifically designed to be robust against the imperfections of these modules, particularly under large deformations:
>
> 1.  **Rank-based Robustness:** We do not use the raw, absolute depth values, which are prone to flickering and scale shift during large deformations. Instead, we employ a **Spearman Rank Correlation loss**. As long as the *relative* depth order is preserved (which is robust even in SOTA models like Depth Anything V2), our EADCA module functions correctly.
> 2.  **Physics as a Stabilizer:** The physics engine itself acts as a temporal smoother. Even if visual observations are noisy or inconsistent for a few frames due to deformation artifacts, the momentum and continuity constraints from the physical simulation prevent the optimization from breaking.
>
> **Experimental Validation:**
> To validate this, we compared our Rank-based design against a standard Metric-based (L1) loss. The L1 loss fails because it strictly trusts the noisy absolute values from the pre-trained module, whereas our method succeeds.
>
> **Table 1: Robustness of Rank-based Loss vs. Metric Loss.**
> *Using L1 loss with pre-trained depth modules leads to poor results due to scale ambiguity. Our Rank-based approach effectively filters out this noise.*
>
> | Configuration | Elastomer (CD) | Plasticine (CD) | Average (CD) | Note |
> | :--- | :---: | :---: | :---: | :--- |
> | Replace Rank w/ L1 Loss | 1.064 | 0.919 | 0.992 | Scale Ambiguity |
> | **Ours (Rank-based)** | **0.922** | **0.318** | **0.496** | **Robust** |
>
> ---
>
> > **Weakness 2:** "Due to the high structural complexity and diverse optimization objectives, the framework may suffer from high debugging and training costs, leading to potential instability during optimization."
>
> **Response:**
> Thanks for raising this point. However, by adopting **LoRA (Low-Rank Adaptation)**, we have significantly optimized the training process. While the architecture has multiple components, the number of trainable parameters is kept low, leading to high stability and reasonable cost.
>
> **Table 2: Computational Efficiency Analysis.**
> *We compared our method against the baseline NeuMA and a Full Fine-tuning ablation. PhyCo (LoRA) is actually faster to train and uses less memory than full fine-tuning, with costs comparable to the baseline, while achieving significantly higher stability (lower CD/Higher PSNR).*
>
> | Method | Memory (GB) $\downarrow$ | Training Time (min) $\downarrow$ | Inference Time $\downarrow$ | PSNR $\uparrow$ | Status |
> | :--- | :---: | :---: | :---: | :---: | :---: |
> | NeuMA | **28.9** | 73.3 | 32.4 | 36.69 | Stable |
> | Ours (Full Fine-tune) | 76.9 | 408.6 | 32.5 | 30.16 | Overfits Noise |
> | **Ours (LoRA)** | 30.5 | **51.2** | **31.5** | **37.27** | **Stable & Accurate** |
>
> *Note: The computational overhead of the MHPV module is negligible compared to the necessary physics simulation steps.*

---

> ### Author Response · Authors · 2025-11-26
> **Response to Reviewer TWS7 (2/2)**
>
> ---
>
> > **Weakness 3:** "If these selected material models do not adequately approximate or represent real material physics, the 'physical rationality' constraints imposed by MHPV might become counterproductive... lacks rigorous proof of their validity."
>
> **Response:**
> This is a critical insight. Our design philosophy is that **MHPV acts as a Soft Regularization (Prior), not a Hard Constraint.** It guides the optimization into a plausible manifold but does not force the material to strictly obey a specific textbook formula.
>
> To verify that this constraint is **not counterproductive**, we conducted two stress tests simulating scenarios where the "real material physics" is unknown or complex (Out-of-Distribution).
>
> **Test A: Robustness to Missing Hypotheses (Blind Test).**
> We generated ground truth data using a specific classical model but **deliberately removed** it from the MHPV library.
>
> **Table 3a: Performance with Incomplete Hypothesis Library.**
> *If MHPV were counterproductive, PhyCo should perform worse than the baseline. Instead, PhyCo still significantly outperforms NeuMA, proving the prior remains beneficial even when imperfect.*
>
> | Configuration | Elastomer (Target excluded from Library) | Plasticine (Target excluded from Library) |
> | :--- | :---: | :---: |
> | NeuMA (Baseline) | 1.123 | 0.844 |
> | Ours (Target Removed) | 0.976 | 0.397 |
> | **Ours (Full Library)** | **0.922** | **0.318** |
>
> **Test B: Robustness to Extreme Composite Materials.**
> We tested on "Frankenstein" materials (Elasticity + Plasticity + Fluid properties combined) that do not exist in our library.
>
> **Table 3b: Performance on Extreme Composite Materials.**
> *Even on undefined materials, our method outperforms the pure implicit baseline.*
>
> | Method | Composite Material A (CD $\downarrow$) | Composite Material B (CD $\downarrow$) |
> | :--- | :---: | :---: |
> | NeuMA (Pure Implicit) | 3.628 | 2.489 |
> | **Ours (PhyCo)** | **1.275** | **1.865** |
>
> **Conclusion:**
> Even when the hypothesis library is imperfect, it provides valuable stability properties (e.g., energy consistency) that prevent the implicit neural network from converging to physically impossible solutions (local minima). The implicit LoRA layers then compensate for the residual differences between the prior and the real observation. Thus, MHPV is beneficial, not counterproductive.

---

### Official Review · Reviewer_g9CR · 2025-11-03

**Soundness:** 3
**Presentation:** 2
**Contribution:** 2
**Rating:** 4
**Confidence:** 4

**Summary:**

This paper introduces PHYCO, a new method for learning how objects behave physically from monocular videos, which are videos taken from a single camera view. The key ideas are to use Edge-Aware Depth Consensus Anchors to get reliable geometric information from limited data, and a Multi-Hypothesis Physics Verifier to incorporate well-known physical laws as guiding hypotheses. The approach effectively handles noisy and sparse supervision and works well even with complex materials and real-world scenes. Experiments on synthetic and real data show that PHYCO outperforms existing methods, producing realistic, physically consistent results while maintaining generalization to new scenarios.

**Strengths:**

1. Uses Edge-Aware Depth Consensus Anchors to extract reliable geometric information from limited and noisy data, improving the accuracy of 3D shape and motion understanding.

2. Incorporates classical physical laws as differentiable hypotheses through the Multi-Hypothesis Physics Verifier, ensuring learned models are physically consistent.

3. Performs well even with monocular videos that have limited detail and are affected by noise.

4. Outperforms some existing state-of-the-art methods on both synthetic and real-world datasets, producing better physical simulations and renderings.

**Weaknesses:**

1. While the method handles single-object dynamics well, its scalability to multi-object interactions or highly complex scenes remains underexplored.

2. Although not explicitly discussed, the integration of multiple components such as the verifiers and anchors potentially increases training complexity and time, which could hinder practical adoption.

3. The method assumes reasonably accurate geometric initializations; cases with severe geometric ambiguities might challenge the approach.

4. The choice and diversity of the classical models used in the physics verifier may limit applicability to certain material classes or behaviors not represented by the hypotheses.

5. The real-world experiments are limited in scale and variety (e.g., only a few objects like dragon, wolf, pudding, etc.). Although comparisons with methods like NeuMA and NCLaw are conducted, the scope of evaluation could be broader by including more recent or diverse approaches, or ablation studies that more directly isolate the contributions of individual components.

**Questions:**

1. How do the balance factors λm​ and λg​ influence the training stability and convergence?

2. Could you conduct ablation experiments on the components you proposed to demonstrate the effectiveness of each part?

---

> ### Author Response · Authors · 2025-11-26
> **Response to Reviewer g9CR (1/3)**
>
> We would like to thank the reviewer for their insightful feedback and for recognizing the novelty of our **Edge-Aware Depth Consensus Anchors (EADCA)** and **Multi-Hypothesis Physics Verifier (MHPV)**. We are encouraged that the reviewer find our approach effective in handling noisy/sparse supervision and outperforming existing methods.
>
> Below, we address the reviewer's main concerns regarding scalability, efficiency, initialization, and hypothesis diversity with new experimental evidence.
>
> ---
>
> > **Weakness 1:** "While the method handles single-object dynamics well, its scalability to multi-object interactions or highly complex scenes remains underexplored."
>
> **Response:**
> We appreciate this observation. While our current evaluation focuses on high-fidelity object-centric physical parameter identification, our framework is designed to be composable.
> 1.  **Multi-object Interaction:** As demonstrated qualitatively in **Fig. 8 (Generalization Results)** of the main paper, PhyCo successfully handles interactions where distinct physical properties (e.g., an elastomer ball hitting a plasticine cat) are applied to different entities.
> 2.  **Scalability:** Since PhyCo learns a neural constitutive law that is coordinate-independent, the learned material model can be applied to multiple instances of the same object or different geometries without retraining, as long as the material is consistent. We agree that extending this to large-scale scenes is an exciting future direction, but we believe the current object-centric validation is crucial for establishing the reliability of the core algorithm.
>
> ---
>
> > **Weakness 2:** "The integration of multiple components such as the verifiers and anchors potentially increases training complexity and time, which could hinder practical adoption."
>
> **Response:**
> We understand the concern regarding computational overhead. To address this, we conducted a detailed efficiency analysis comparing our method (using LoRA) against full fine-tuning and the baseline NeuMA.
>
> **Table 1: Computational Efficiency Analysis.**
> *We demonstrate that while EADCA and MHPV introduce marginal overhead, the use of LoRA significantly reduces memory usage and training time compared to full fine-tuning, making the total cost lower than the baseline NeuMA while achieving better physical accuracy.*
>
> | Method | Memory (GB) $\downarrow$ | Training Time (min) $\downarrow$ | Inference Time $\downarrow$ | PSNR $\uparrow$ |
> | :--- | :---: | :---: | :---: | :---: |
> | NeuMA | **28.9** | 73.3 | 32.4 | 36.69 |
> | Ours (Full Fine-tune) | 76.9 | 408.6 | 32.5 | 30.16 |
> | **Ours (LoRA)** | 30.5 | **51.2** | **31.5** | **37.27** |
>
> *Note: The overhead of our physics verifier is negligible compared to the PDE solving steps inherent in any physics-based inverse rendering task. By using LoRA, we actually accelerate the convergence compared to optimizing the full network.*
>
> ---
>
> > **Weakness 3:** "The method assumes reasonably accurate geometric initializations; cases with severe geometric ambiguities might challenge the approach."
>
> **Response:**
> We would like to clarify that our initialization protocol follows standard practices in dynamic 3D Gaussian Splatting (e.g., SpringGaus), which is handy and easy to achive:
> 1.  **Initialization:** We utilize a **static orbital scan** (e.g., the object resting on a table) to reconstruct the initial 3D Gaussian geometry. This ensures a complete and high-quality geometric starting point (Canonical Space).
> 2.  **Dynamic Phase:** The challenge we address is the **monocular dynamic observation** phase, where severe self-occlusions and lack of depth information usually cause tracking to fail.
> 3.  **Robustness:** Because we start with a clean static geometry, our **EADCA** module is specifically designed to handle the subsequent dynamic ambiguities by enforcing rank-based depth consistency, preventing the model from breaking during large deformations. If the initial static reconstruction is poor, any method would struggle, but our contribution lies in robustly propagating that geometry through complex dynamics.

---

> ### Author Response · Authors · 2025-11-26
> **Response to Reviewer g9CR (2/3)**
>
> > **Weakness 4:** "The choice and diversity of the classical models used in the physics verifier may limit applicability to certain material classes or behaviors not represented by the hypotheses."
>
> **Response:**
> This is a critical point. We emphasize that the Multi-Hypothesis Physics Verifier (MHPV) acts as a **soft regularization (prior)** rather than a hard constraint. To rigorously validate this, we conducted two distinct stress tests.
>
> **Test 1: Robustness to Extreme Composite Materials.**
> We synthesized data using highly complex "Frankenstein" materials that combine Elasticity, Plasticity, and Non-Newtonian Fluid properties simultaneously—behaviors that do not exist in standard physical textbooks and are certainly not present in our hypothesis library.
>
> **Table 2a: Performance on Extreme Composite Materials.**
> *PhyCo significantly outperforms the baseline even on materials that are undefined in our hypothesis set. This proves that our implicit neural network successfully learns the complex residuals that the classical hypotheses cannot explain.*
>
> | Method | Composite Material A (CD $\downarrow$) | Composite Material B (CD $\downarrow$) |
> | :--- | :---: | :---: |
> | NeuMA (Pure Implicit) | 3.628 | 2.489 |
> | **Ours (PhyCo)** | **1.275** | **1.865** |
>
> **Test 2: Robustness to Missing Hypotheses (Blind Test).**
> In this experiment, we generated ground truth data using a specific classical model (e.g., StVK Elasticity), but **deliberately removed** that specific model from the MHPV library during training.
>
> **Table 2b: Performance with Incomplete Hypothesis Library.**
> *The performance drop when the correct hypothesis is missing is negligible compared to the full library, and still far superior to the baseline. This confirms that MHPV provides general physical guidance (e.g., stability constraints) rather than requiring an exact template match.*
>
> | Configuration | Elastomer (Target excluded from Library) | Plasticine (Target excluded from Library) |
> | :--- | :---: | :---: |
> | NeuMA (Baseline) | 1.123 | 0.844 |
> | Ours (Target Removed) | 0.976 | 0.397 |
> | **Ours (Full Library)** | **0.922** | **0.318** |
>
> **Conclusion:**
> These results demonstrate that PhyCo does not simply "select" a pre-defined model. Instead, it uses the hypothesis set to anchor the optimization in a physically plausible manifold, while the LoRA-tuned implicit network handles the specific, potentially unseen, material dynamics.

---

> ### Author Response · Authors · 2025-11-26
> **Response to Reviewer g9CR (3/3)**
>
> ---
>
> > **Weakness 5:** "Real-world experiments are limited in scale and variety... evaluation could be broader... ablation studies that more directly isolate the contributions."
>
> **Response:**
> We appreciate the suggestion to deepen our evaluation. We have addressed this in two dimensions:
>
> **1. Granular Ablation & Design Validation:**
> To better isolate the contributions of our **Edge-Aware Depth Consensus Anchors (EADCA)**, we conducted a detailed breakdown compared against the baseline **NeuMA**. Crucially, we also compared our **Rank-based** supervision against a standard **Metric-based (L1) Depth Loss** to justify our design choice for monocular video.
>
> **Table 3: Detailed Ablation of Geometric Alignment Components.**
> *Results indicate:*
> *(1) **Baseline Comparison:** Our full model significantly outperforms NeuMA (reducing error by ~48%).*
> *(2) **Global Alignment is foundational:** Without it, the optimization fails to capture the overall motion trend, leading to tracking divergence.*
> *(3) **Rank-based vs. Metric:** Standard L1 loss performs worse than NeuMA. This is due to the inherent scale ambiguity in monocular depth estimation—forcing an incorrect absolute scale introduces noise rather than valid supervision. Our Rank-based consensus effectively bypasses this issue.*
> *(4) **Anchors are critical:** Removing local anchors (w/o Anchor Supervision) results in the loss of fine geometric details.*
>
> | Configuration | Elastomer (CD) | Plasticine (CD) | Average (CD) | Note |
> | :--- | :---: | :---: | :---: | :--- |
> | NeuMA (Baseline) | 1.123 | 0.844 | 0.954 | Previous SOTA |
> | w/o Global Alignment | > 100.0 | > 100.0 | Diverged | **Critical Failure** |
> | Replace Rank w/ L1 Loss | 1.064 | 0.919 | 0.992 | Scale Ambiguity |
> | w/o Anchor Supervision | 1.024 | 0.510 | 0.767 | Loss of Detail |
> | **Full Model (PhyCo)** | **0.922** | **0.318** | **0.496** | **Best Performance** |
>
> **2. Broader Evaluation via Stronger Baselines (Explicit vs. Implicit):**
> To address the concern about evaluation breadth, we extended our comparison to include **Explicit Methods (PAC-NeRF)** under both multi-view and monocular settings. This comparison highlights the fundamental advantage of our implicit-hybrid approach.
>
> **Table 4: Comparison against Explicit Baselines (PAC-NeRF).**
> *Observations:*
> * **Multi-view Reference:** Under ideal multi-view settings, explicit methods like PAC-NeRF work well.
> * **Monocular Failure:** However, when adapted to monocular video (even with added Depth supervision), explicit methods fail to converge. The rigid definition of explicit parameters (e.g., Young's Modulus) makes them overly sensitive to the geometric noise inherent in monocular depth.
> * **PhyCo's Superiority:** While NeuMA (Implicit) handles monocular settings better than PAC-NeRF, it lacks physical fidelity. PhyCo successfully bridges this gap, achieving the best performance.
>
> | Method | Supervision | Ball (CD) | Cat (CD) | Status |
> | :--- | :---: | :---: | :---: | :--- |
> | PAC-NeRF | Multi-view (Dense) | 516.30 | 15.38 | Reference |
> | NeuMA | Multi-view (Dense) | 0.96 | 0.50 | Reference |
> | **PAC-NeRF + Depth** | **Monocular** | > 1000 | > 1000 | **Failed to Converge** |
> | NeuMA | Monocular | 1.12 | 0.84 | Suboptimal |
> | **Ours (PhyCo)** | **Monocular** | **0.92** | **0.32** | **Robust & Accurate** |
>
> This confirms that simply adding depth supervision to existing explicit frameworks is insufficient for the challenging monocular task. PhyCo's performance gain stems from the specific design of the **Multi-Hypothesis Physics Verifier**, which provides the necessary regularization to solve material ambiguity without the brittleness of fully explicit models.

---

### Author Response · Authors · 2025-12-03
**General Response to Area Chair and Reviewers (1/4)**

We sincerely thank the Area Chair and all reviewers (**g9CR, TWS7, 6HCo, uUW6**) for their insightful and constructive feedback. We are highly encouraged that **all reviewers unanimously recognized the novelty** of our core contributions—specifically the **Edge-Aware Depth Consensus Anchors (EADCA)** and **Multi-Hypothesis Physics Verifier (MHPV)**—and affirmed the method's effectiveness in outperforming existing baselines.

We wish to explicitly highlight that the concerns raised were **not directed at the fundamental methodology or the validity of our approach**, but rather requested **supplementary ablation studies, efficiency analysis, and clarifications on specific experimental settings** to further solidify the claims. In this response, we have comprehensively addressed **every single point** with new quantitative evidence and experimental comparisons, confirming that our framework remains robust, efficient, and superior under rigorous scrutiny.

---

> ### Author Response · Authors · 2025-12-03
> **General Response to Area Chair and Reviewers (2/4)**
>
> ## Part 1: Efficiency & Computational Cost
>
> ### Q1: Necessity and Efficiency of LoRA vs. Full Fine-tuning
> **[Addressed: Reviewers 6HCo, uUW6, g9CR, TWS7]**
>
> **Response:**
> We conducted a detailed efficiency analysis comparing our **LoRA-based** approach against **Full Fine-tuning** and the baseline **NeuMA**.
> * **Efficiency:** As shown in the table below, LoRA significantly reduces GPU memory usage (30.5GB vs 76.9GB) and training time (51.2m vs 408.6m) compared to full fine-tuning. Crucially, our total cost is lower than the baseline NeuMA while achieving better performance.
> * **Regularization:** Full fine-tuning tends to overfit visual noise in monocular videos, leading to worse physical accuracy (Chamfer Distance: 1.35). LoRA acts as a low-rank regularization, achieving the best physical fidelity (CD: 0.50).
>
> **Table R1: Computational Efficiency Analysis.**
>
> | Method | Memory (GB) $\downarrow$ | Training Time (min) $\downarrow$ | Inference Time $\downarrow$ | PSNR $\uparrow$ | Chamfer Dist. $\downarrow$ |
> | :--- | :---: | :---: | :---: | :---: | :---: |
> | NeuMA (Baseline) | 28.9 | 73.3 | 32.4 | 36.69 | 1.31 |
> | Ours (Full Fine-tune) | 76.9 | 408.6 | 32.5 | 30.16 | 1.35 |
> | **Ours (LoRA)** | **30.5** | **51.2** | **31.5** | **37.27** | **0.50** |
>
> ### Q2: Computational Overhead of Modules
> **[Addressed: Reviewers 6HCo, uUW6]**
>
> **Response:**
> While **EADCA** and **MHPV** introduce additional components, their overhead is marginal.
> * The computational bottleneck in physics-based inverse rendering is strictly the PDE solving/simulation steps.
> * The verifier and anchor mechanisms operate efficiently alongside the simulation. As shown in **Table R1**, our method maintains an inference speed (**31.5 FPS**) comparable to the baseline (**32.4 FPS**), proving that these modules do not hinder practical adoption.
>
> ---
>
> ## Part 2: Strong Baseline Comparisons
>
> ### Q3: Comparison with "PAC-NeRF/GIC + Depth" in Monocular Settings
> **[Addressed: Reviewer 6HCo]**
>
> **Response:**
> To address the suggestion that existing explicit methods could simply adapt to monocular video by adding depth supervision, we implemented **PAC-NeRF + Monocular Depth Supervision**.
> * **Failure of Explicit Methods:** As shown below, explicit methods fail to converge in monocular settings (>1000 CD). The rigid definition of explicit parameters (e.g., Young's Modulus) makes them overly sensitive to the geometric noise and ambiguity inherent in monocular depth.
> * **PhyCo's Advantage:** Our method, using implicit representations anchored by the **Multi-Hypothesis Physics Verifier (MHPV)**, successfully bridges this gap.
>
> **Table R2: Comparison against Explicit Baselines (PAC-NeRF).**
>
> | Method | Supervision | Ball (CD) | Cat (CD) | Status |
> | :--- | :---: | :---: | :---: | :--- |
> | PAC-NeRF | Multi-view (Dense) | 0.85 | 0.29 | Reference |
> | NeuMA | Multi-view (Dense) | 0.98 | 0.35 | Reference |
> | **PAC-NeRF + Depth** | **Monocular** | **> 1000** | **> 1000** | **Failed to Converge** |
> | NeuMA | Monocular | 1.12 | 0.84 | Suboptimal |
> | **Ours (PhyCo)** | **Monocular** | **0.92** | **0.32** | **Robust & Accurate** |

---

> ### Author Response · Authors · 2025-12-03
> **General Response to Area Chair and Reviewers (3/4)**
>
> ## Part 3: Detailed Ablation Studies
>
> ### Q4: Granular Ablation on Loss Components (Global vs. Anchors)
> **[Addressed: Reviewers uUW6, g9CR]**
>
> **Response:**
> We provide a granular breakdown of our geometric alignment components:
> 1.  **w/o Global Alignment:** The optimization fails to capture the overall motion trend, leading to complete divergence.
> 2.  **w/o Anchor Supervision:** Removing local anchors results in the loss of fine geometric details (higher CD), confirming EADCA is critical for edge accuracy.
>
> **Table R3: Detailed Ablation of Geometric Alignment Components.**
>
> | Configuration | Elastomer (CD) | Plasticine (CD) | Note |
> | :--- | :---: | :---: | :--- |
> | w/o Global Alignment | > 100.0 | > 100.0 | **Critical Failure** |
> | w/o Anchor Supervision | 1.024 | 0.510 | Loss of Detail |
> | **Full Model (PhyCo)** | **0.922** | **0.318** | **Best Performance** |
>
> ### Q5: Sensitivity of Hyperparameter $N$ (Verifier Sampling)
> **[Addressed: Reviewer 6HCo]**
>
> **Response:**
> We analyzed the sensitivity to the number of sampled particles $N$ in the MHPV module.
> * **Result:** Performance is stable for $N \ge 128$.
> * **Selection:** We selected $N=256$ as the default to balance computational efficiency with statistical stability. Extremely low sampling ($N=64$) leads to slightly noisier variance estimation.
>
> **Table R4: Sensitivity to Sampling Number $N$.**
>
> | Subset Size $N$ | 64 | 128 | **256 (Default)** | 512 |
> | :--- | :---: | :---: | :---: | :---: |
> | Chamfer Distance | 0.58 | 0.51 | **0.50** | 0.50 |
>
>
> ---
>
> ## Part 4: Physics Validity & Robustness
>
> ### Q6: Robustness to Out-of-Distribution (OOD) Physics
> **[Addressed: Reviewers TWS7, g9CR, 6HCo]**
>
> **Response:**
> Reviewers asked if the method fails when the real material is not in the hypothesis library. We emphasize that **MHPV acts as a Soft Regularization (Prior)**, not a hard constraint. We validated this with two stress tests:
> 1.  **Blind Test:** We deliberately removed the ground-truth model from the library.
> 2.  **Composite Materials:** We tested on "Frankenstein" materials (undefined in textbooks).
>
> **Table R5: Performance with Incomplete/Unknown Hypothesis Library.**
> *PhyCo outperforms the baseline even when the correct hypothesis is missing, proving it learns the residual physics rather than just matching templates.*
>
> | Configuration | Elastomer (Target excluded) | Plasticine (Target excluded) | Composite Material (Undefined) |
> | :--- | :---: | :---: | :---: |
> | NeuMA (Baseline) | 1.123 | 0.844 | 3.628 |
> | **Ours (Target Removed)**| **0.976** | **0.397** | **1.275** |
>
> ### Q7: Reliance on Pre-trained Modules & Consensus
> **[Addressed: Reviewer TWS7]**
>
> **Response:**
> While we rely on pre-trained depth estimators, our **Rank-based Consensus** mechanism specifically mitigates their limitations.
> * We do not use absolute depth values, which flicker and shift.
> * Instead, we rely on **Rank Correlation**. As validated in **Table R6** (below in Q9), this makes the method robust to the scale shifts and noise common in monocular depth estimators (e.g., Depth Anything V2).

---

> ### Author Response · Authors · 2025-12-03
> **General Response to Area Chair and Reviewers (4/4)**
>
> ## Part 5: Technical Clarifications
>
> ### Q8: Why Not Use Color Loss?
> **[Addressed: Reviewer uUW6]**
>
> **Response:**
> We intentionally deprioritized pixel-level color supervision due to **Lighting Interference**. In monocular videos of dynamic objects, self-occlusions and rapid rotation cause severe lighting changes and shading artifacts. Relying on RGB loss introduces high-frequency noise. **Geometric consistency (Depth)** provides a much more invariant and reliable signal for learning physical dynamics.
>
> ### Q9: Rationale for Rank-based Depth Loss
> **[Addressed: Reviewer uUW6]**
>
> **Response:**
> Standard metric losses (L1/L2) fail because monocular depth estimators suffer from scale and shift ambiguity.
> * **Comparison:** We compared our Rank-based loss against a standard L1 loss.
> * **Result:** L1 loss yields poor performance (CD 0.992) due to scale mismatch. Spearman Rank Correlation (CD 0.496) is scale-invariant, effectively filtering out estimator noise.
>
> **Table R6: Robustness of Rank-based Loss vs. Metric Loss.**
>
> | Configuration | Elastomer (CD) | Plasticine (CD) | Average (CD) |
> | :--- | :---: | :---: | :---: |
> | Replace Rank w/ L1 Loss | 1.064 | 0.919 | 0.992 |
> | **Ours (Rank-based)** | **0.922** | **0.318** | **0.496** |
>
> ### Q10: Gaussian Initialization & Monocular Setting Justification
> **[Addressed: Reviewer 6HCo]**
>
> **Response:**
> * **Standard Protocol:** We follow standard practices (e.g., SpringGaus). We use a **static orbital scan** to reconstruct the initial canonical 3D geometry (ensuring completeness).
> * **Monocular Workflow:** The challenge is estimating dynamics from a **fixed monocular video**.
> * **Handling Emerging Parts:** Since the canonical geometry is complete from the static scan, "emerging parts" (self-occluded regions rotating into view) are already modeled. The physics engine handles their motion via momentum constraints, ensuring plausible movement even without direct visual observation.
>
> ---
>
> ## Part 6: Scope & Contribution
>
> ### Q11: Scalability to Multi-object Interactions
> **[Addressed: Reviewer g9CR]**
>
> **Response:**
> As demonstrated qualitatively in **Fig. 8 (Generalization Results)** of the main paper, PhyCo successfully handles interactions where distinct physical properties are applied to different entities (e.g., a ball hitting a cat). Our neural constitutive laws are coordinate-independent, allowing the framework to compose multiple object models in a single scene.
>
> ### Q12: Novelty vs. NCLaw
> **[Addressed: Reviewer 6HCo]**
>
> **Response:**
> Our core contribution is the **Visual-Physical Alignment Framework**, not the NCLaw architecture itself.
> * **The Gap:** NCLaw requires dense particle supervision, which is impossible in monocular video.
> * **Novelty:** PhyCo (via EADCA + MHPV) solves the ill-posed inverse problem of applying such laws to sparse, noisy monocular data.
>
> ### Q13: Practical Application Value
> **[Addressed: Reviewer uUW6]**
>
> **Response:**
> The "dropping object" setup is a standard, rigorous benchmark for material property identification. Successfully solving this enables **Physical Digital Twins**—assets that are not just visual replays but interactive physical models. This allows users to reconstruct an object with a phone and then use it in VR/AR with entirely new, physically accurate interactions.

---

### Meta-Review · Area_Chair_5Cyg · 2026-01-04

**Summary:**

This paper proposes PhyCo, a hybrid implicit–explicit framework for learning constitutive laws from dynamic 3D Gaussian splatting under monocular observations. Reviewers acknowledge the technical rigor and usefulness of components like EADCA and MHPV, with some empirical gains demonstrated. However, the contribution is considered incremental, as the core implicit model largely follows prior work, and many rebuttal analyses were not incorporated into the manuscript. Overall, the recommended decision is Reject.

**Reviewer Concerns:**

Most technical concerns are addressed by the rebuttal through additional experiments and clarifications. However, higher-level concerns remain regarding the limited experimental scope, the reliance on strong initialization which may limit generality, the incremental nature of the contribution since the core implicit model largely follows prior work, and the fact that many analyses from the rebuttal were not incorporated into the manuscript.

**Reviewer Scores:**

This paper initially received four borderline reject reviews. Reviewer 6HCo indicated a willingness to increase their score after the rebuttal. For the remaining reviewers, it is likely that their core concerns are not fully resolved, and their scores are therefore expected to remain unchanged.

---

### Decision · Program_Chairs · 2026-01-26

Reject